# Sem-Detect: Semantic Level Detection of AI Generated Peer-Reviews

André V. Duarte [1 2]   Brian Tufts [1]   Aditya Oke [1]   Fei Fang [1]   Arlindo L. Oliveira [2]   Lei Li [1]

## Abstract

*How can we distinguish whether a peer review was written by a human or generated by an AI model?* We argue that, in this setting, authorship should not be attributed solely from the textual features of a review, but also from the ideas, judgments, and claims it expresses. To this end, we propose Sem-Detect, an authorship detection method for peer reviews that operationalizes this principle by combining textual features with claim-level semantic analysis. Sem-Detect compares a target review against multiple AI-generated reviews of the same paper, leveraging the observation that different AI models tend to converge on similar points, while human reviewers introduce more unique and diverse ones. As a result, Sem-Detect is able to distinguish fully AI reviews from authentic human-written ones, including those that have been refined using an LLM but still reflect human judgment. Across a dataset of over 20,000 peer reviews from ICLR and NeurIPS conferences, Sem-Detect improves over the strongest baseline by 25.5% in TPR@0.1% FPR in the binary setting. Moreover, in the three-class scenario, we empirically show that LLM refinement preserves the semantic signals of human reviews, which remain distinct from the patterns exhibited by fully AI-generated text; as a result, fewer than 3.5% of LLM-refined human reviews are misclassified as AI-generated.

## 1. Introduction

Peer review is fundamental to scientific progress. When researchers submit a paper, they expect substantive feedback from domain experts; feedback that can clarify the work for future readers and guide authors in strengthening their contributions. However, with the rapid advancement of

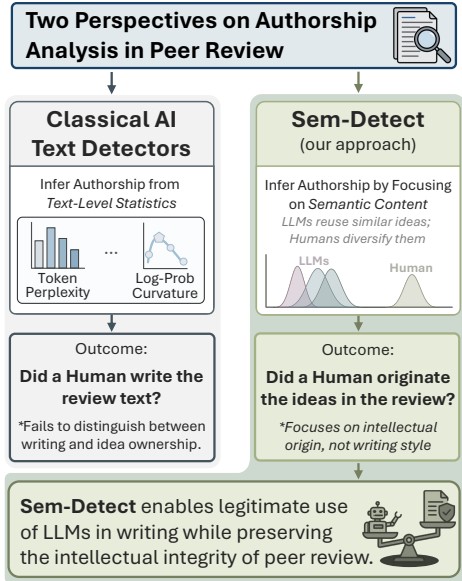

*Figure 1.* Classical AI-text detectors rely on textual features to decide whether a review was written by a human. Sem-Detect instead infers authorship by leveraging the semantic content of expressed ideas, thereby distinguishing fully AI-generated reviews from LLM-refined human ones.

large language models (LLMs), there is growing evidence of AI-generated content appearing in peer reviews (Liang et al., 2024; Zhou et al., 2025). This trend raises a serious concern: authors may no longer know whether the feedback they receive reflects genuine human judgment.

While initial responses from the research community were strict, as exemplified by ICML 2025's ban on any use of LLMs in the review process (ICML Conference Chairs, 2025), there has since been a notable policy shift. ICML 2026 now allows LLM assistance for editing and improving the clarity of reviews (ICML Conference Chairs, 2026). This shift reflects a recognition that the appropriate boundary lies not in whether an LLM touched the text, but in whether the expressed ideas originated from a human or from a machine. A reviewer who drafts an assessment and later uses an LLM to improve its readability is engaging in a qualitatively different activity than one who prompts an LLM to generate an entire review. Detecting this distinction, however, poses a technical challenge that existing methods are not well-equipped to address (Fitzgibbon et al., 2024).

[1]Language Technologies Institute, Carnegie Mellon University [2]INESC-ID, Instituto Superior Técnico, ULisboa. Correspondence to: André V. Duarte <aduarte@andrew.cmu.edu>.

*Proceedings of the 43rd International Conference on Machine Learning*, Seoul, South Korea. PMLR 306, 2026. Copyright 2026 by the author(s).

Current approaches to AI-text detection can be broadly organized along two axes: (i) general-purpose methods designed to work across diverse domains, and (ii) domain-specific methods tailored to particular contexts such as peer review.

General-purpose methods can range from zero-shot statistical approaches such as FastDetectGPT (Bao et al., 2024), which leverage text conditional probability curvature to identify machine-generated content, to more sophisticated techniques like RADAR (Hu et al., 2023), which use adversarial training to achieve increased robustness against LLM-based paraphrasing. However, because these approaches rely on surface-level textual signals, when applied to the peer-review domain they struggle to distinguish human-authored judgments that have been linguistically refined by an LLM from content generated end-to-end by an LLM.

Domain-specific methods, by contrast, leverage contextual information unique to the task. For example, Yu et al. (2026) generate synthetic AI reviews from research papers and train Anchor, which embeds entire reviews and compares them to a reference AI review using cosine similarity to infer authorship. However, operating at the full-review level limits interpretability, making it difficult to identify which claims drive a given classification.

To address these limitations while building on the strengths of existing approaches, we propose Sem-Detect. Like general-purpose methods, Sem-Detect extracts textual features from the target review, as these remain fundamental for distinguishing purely human text from fully AI-generated content. However, inspired by domain-specific approaches such as Anchor (Yu et al., 2026), Sem-Detect moves beyond text-level analysis by explicitly modeling the semantic content of reviews. Rather than embedding entire reviews and comparing them as a whole, our method operates at the claim level: it pairs each target review with multiple AI-generated reviews of the same paper and measures semantic similarity at a finer granularity. This design exploits the observation that different AI models tend to converge on similar points when reviewing the same paper, while human reviewers introduce more unique judgments. As a result, we can distinguish not only between human and AI authorship, but also identify cases in which a human assessment has been refined by an LLM, treating such reviews as a separate class rather than mixing them with fully AI-generated text.

Using a corpus of over 20,000 reviews (human-written, LLM-refined, and AI-generated) constructed from 800 papers across ICLR and NeurIPS conferences, we train and evaluate Sem-Detect. Human reviews collected up to 2022 serve as clean baselines. To assess robustness beyond these controlled conditions, we further evaluate the method on: AI-generated reviews produced by unseen models and prompting strategies; cross-domain reviews from a medical imaging venue; and recent submissions from ICLR 2026.

Our main contributions are as follows:

- We identify a consistent pattern in peer reviews: when reviewing the same paper, AI-generated reviews exhibit higher claim-level overlap with one another than human-written reviews, including those refined using LLMs.

- We operationalize this insight in Sem-Detect, a practical detection framework that combines textual features with claim-level semantic analysis to distinguish human-written, LLM-refined, and fully AI-generated reviews.

- We construct and release a dataset of over 20,000 peer reviews spanning human-written, AI-generated, and LLM-refined variants from ICLR and NeurIPS (pre-2022), with additional evaluation data from a medical imaging venue and ICLR 2026.

- Experiments show that Sem-Detect improves over the strongest prior detector by 25.5% in TPR@0.1% FPR in binary detection, with fewer than 3.5% of LLM-refined human reviews misclassified as AI-generated. We further validate robustness to unseen models, cross-domain transfer, and temporal generalization.

## 2. Related Work

Detecting machine-generated text has become a central challenge in the NLP community, with methods spanning watermarking, zero-shot detection, and supervised classification (Jawahar et al., 2020; Ghosal et al., 2023; Wu et al., 2025; Rao et al., 2025). We organize prior work along two axes: general-purpose methods designed for broad applicability, and domain-specific approaches for peer review.

### 2.1. General-Purpose AI-Text Detection

**Watermarking.** Watermarking embeds detectable statistical signals during text generation, with some methods offering provable guarantees on false positive rates (Kirchenbauer et al., 2023; Zhao et al., 2024). However, watermarking requires control over the generation process and therefore has limited applicability in settings where the source model is unknown.

**Zero-shot methods.** Zero-shot detectors operate without task-specific training data by exploiting statistical properties of LLM outputs (Hans et al., 2024). DetectGPT (Mitchell et al., 2023) introduced the concept of probability curvature, observing that perturbations of LLM-generated text tend to reduce its log-probability in the source model. In contrast, human-written text does not exhibit the same systematic behavior. Follow-up work such as Fast-DetectGPT (Bao et al., 2024) achieves comparable accuracy with reduced computational cost. Other approaches rely on simpler statistical metrics, including perplexity (Gutiérrez Megías et al., 2024) and entropy (Lavergne et al., 2008).

**Trained detectors.** Supervised methods train classifiers on human and AI-generated text. Early approaches fine-tuned models like RoBERTa (Liu et al., 2019) on detection datasets (Zellers et al., 2019; Solaiman et al., 2019), but these methods are often sensitive to adversarial scenarios such as LLM-based paraphrasing. To address this, recent work like RADAR (Hu et al., 2023) jointly trains a detector and a paraphraser in an adversarial framework, where the paraphraser learns to generate evasive rewrites while the detector learns to remain robust against them. However, even robust trained detectors operate solely on the target text, without access to contextual information (e.g., the manuscript under review) that could provide additional discriminative signal.

## 2.2. Domain-Specific Detection in Peer Review

While general-purpose detectors focus only on the target text, peer review methods can exploit the relationship between reviews and manuscripts, as well as the structured nature of review writing.

**Leveraging domain signals.** Liang et al. (2024) provided early evidence of LLM-generated content in peer reviews by tracking the surge of adjectives characteristic of Chat-GPT (OpenAI, 2022) outputs. Building on this, the Term Frequency (TF) model introduced by Kumar et al. (2024) exploits repetitive token usage patterns in AI-generated text and demonstrates that even simple domain-tailored signals can outperform more generic detection strategies.

**Manuscript-conditioned detection.** Anchor (Yu et al., 2026) conditions detection on the paper under review. The method generates a synthetic AI review for the target paper and compares it with the candidate review using embedding-based cosine similarity: reviews that closely resemble the AI reference are flagged as machine-generated. However, Anchor operates at the full-review level, embedding entire reviews as single vectors, limiting the method's ability to disentangle partial semantic overlap from end-to-end AI authorship. In a complementary direction, Rao et al. (2025) embed hidden instructions in submitted PDFs that induce LLMs to insert detectable watermarks into generated reviews. However, this requires venue-level adoption, which limits practical deployment.

**Beyond binary detection.** Most recently, EditLens (Thai et al., 2026) re-frames the task by moving beyond binary classification to quantify the extent of AI editing on a continuous scale. This represents an important conceptual shift, acknowledging that the boundary between human and AI authorship is not always sharp. However, EditLens focuses on estimating edit intensity rather than distinguishing the origin of the underlying ideas. As a consequence, a human review

fully polished by an LLM and an AI-generated review may receive similar scores, despite representing fundamentally different authorship scenarios.

## 2.3. Granularity in Semantic Comparison

Our approach is inspired by work in the retrieval literature showing that the granularity of text representation has a strong impact on downstream performance. Dense X Retrieval (Chen et al., 2024) adopts atomic propositions as retrieval units, ensuring that each representation corresponds to a single, semantically independent claim. Similarly, LumberChunker (Duarte et al., 2024) shows that segmenting text along semantic boundaries is more effective than arbitrary chunking strategies. Together, these findings highlight a common principle: large document-level representations mix multiple semantic units, which reduces precision in similarity-based comparison. For the same reason, Sem-Detect operates at the claim level, allowing us to better isolate the semantic patterns that distinguish AI-generated content from human-written reviews.

## 3. Sem-Detect

Sem-Detect addresses the problem of peer-review authorship attribution by distinguishing between fully human-written reviews, human reviews refined by an LLM, and end-to-end machine generated ones. As illustrated in Figure 2, the pipeline consists of two main stages: (i) the construction of a peer-review dataset spanning these three classes, and (ii) the extraction of textual and claim-level semantic features from this data to train a detection model. We describe the key design choices of each stage below. Further details are provided in Appendices A.1-A.5.

### 3.1. Training Data Construction

**Human reviews.** We randomly sample 200 papers from each of ICLR and NeurIPS for the years 2021 and 2022, resulting in a total of 800 papers. We crawl both papers and their associated reviews from OpenReview,[1] retrieving the blind submission version for each paper to ensure consistency with what reviewers saw at the time of writing. In total we have 3,065 human-written reviews.

**Fully AI-generated reviews.** Using the sampled papers, we generate a set of fully AI-written reviews. While every conference has their own reviewing guidelines, peer reviews across venues generally follow a common structure consisting of: (1) a summary of the paper, (2) a discussion of strengths, (3) a discussion of weaknesses, and (4) clarification questions for the authors. We leverage this structure to prompt four different LLMs to generate their reviews.

---

[1] https://openreview.net

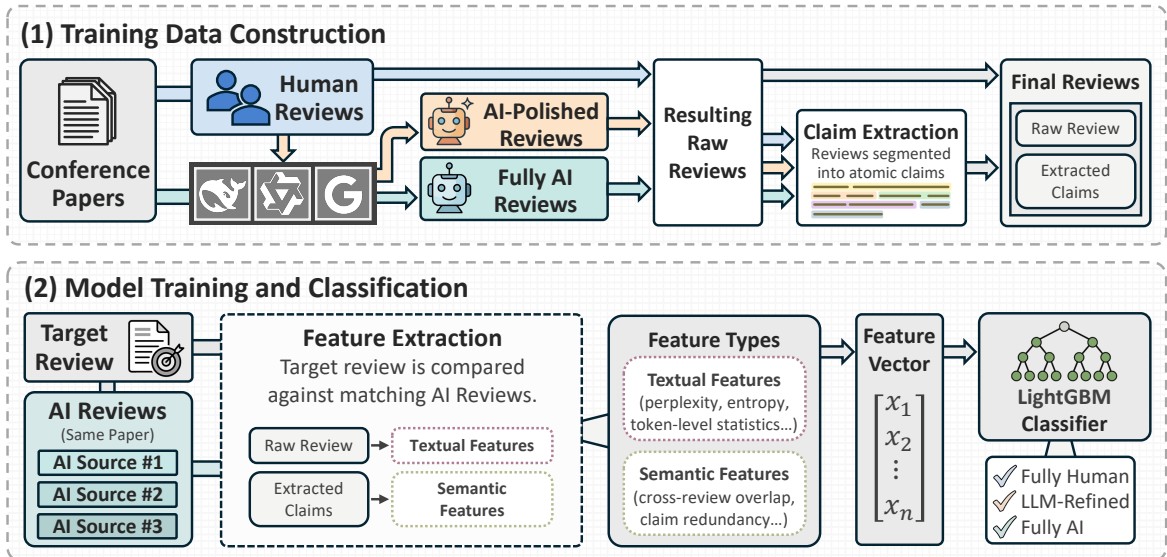

*Figure 2.* Sem-Detect pipeline. We construct our dataset by prompting LLMs to generate fully AI reviews from conference papers and to refine authentic human reviews, creating three classes. For classification, each target review (from any class) is paired with multiple AI-generated reference reviews of the same paper. We extract textual features from the target review and semantic features from the target-reference comparisons. These combined features train a LightGBM classifier to distinguish between human-written, LLM-refined, and fully AI-generated reviews.

A second consideration concerns the distribution of review scores. To avoid the optimism bias documented in Russo et al. (2025), we explicitly specify the target score during generation. As such, for each paper, LLMs generate reviews corresponding to the distinct scores assigned by human reviewers, ensuring balanced coverage of evaluation outcomes, and resulting in a total of 6,768 AI-generated reviews.

**LLM-refined reviews.** In contrast to fully AI-generated reviews, this class originates from human-written assessments. It reflects the realistic scenario in which a reviewer drafts an initial evaluation and subsequently uses an LLM to improve its clarity. As such, during this refinement step, the LLM is explicitly instructed to preserve all original judgments and to avoid introducing new content. This procedure is applied to each human review using the four LLMs, and results in 12,332 LLM-refined reviews.

**Post-processing.** Both fully AI-generated and LLM-refined reviews can include elements that directly reveal how they were produced, such as sentences like "Here is the review of ...". We use an LLM to remove these artifacts through a post-processing step, resulting in plain-text reviews that follow the same format as human ones.

**Claim extraction.** A central premise of Sem-Detect is that authorship signals are reflected not only in writing style, but also in the content of a review. To capture this information, we use an LLM to extract structured claim-level representations from each text. Specifically, we semantically segment

each review into bullet points belonging to five categories: factual restatement, evaluation, constructive input, clarification dialogue, and meta-commentary. Each bullet point is designed to capture a single claim while preserving the reviewer's original phrasing whenever possible.

### 3.2. Model Training and Classification

Let $t$ denote a target review and let $p$ be the paper it evaluates. We assume access to a set of AI-generated reference reviews $\mathcal{A}_p = \{a_1, \ldots, a_k\}$ for the same paper, produced by prompting $k$ different LLMs. Our goal is to learn a function $f(t, \mathcal{A}_p) \to \{0, 1, 2\}$ that maps the target review and its references to one of three classes: human-written, LLM-refined, or fully AI-generated. Additional details are reported in Appendices B.1–B.6.

**Reference review pairing.** For each target review $t$, we pair it with $k = 3$ AI-generated reference reviews of the same paper. Reference reviews are selected under two conditions: (i) they share the same evaluation score as $t$, so that semantic comparisons are not affected by differences in overall judgment; and (ii) when $t$ is AI-generated, they are produced by different models, to avoid inflated similarity scores from model-specific patterns (Xu et al., 2024).

**Claim filtering and embedding** As described in Section 3.1, each review is segmented into five claim categories, but only a subset is informative for authorship attribution. For semantic analysis, we consider only claims from cate-

gories that reflect evaluative judgment, namely (i) evaluation, (ii) constructive input, and (iii) clarification dialogue.

**Feature extraction and classifier training.** For each target review $t$, we extract a nine-dimensional feature vector comprising five semantic features and four textual features. Semantic features are computed from claim embeddings and their comparisons to AI-generated reference reviews, while textual features come directly from the raw text of $t$.

Let $\mathcal{C}_t = \{c_1, \ldots, c_n\}$ denote the set of claims extracted from $t$, and let $\mathcal{A}_p = \{a_1, \ldots, a_k\}$ denote the set of AI-generated reference reviews for the same paper. For each target claim $c_i$ and each reference review $a_j$, with claim set $\mathcal{C}_{a_j}$, we compute the best-match similarity

$$s_{i,j} = \max_{c \in \mathcal{C}_{a_j}} \cos(\phi(c_i), \phi(c)),$$

where $\phi(\cdot)$ denotes a claim embedding function. We further define $s_i = \max_j s_{i,j}$ as the best-match similarity of $c_i$ across all reference reviews.

Semantic features include: (i) the proportion of target claims whose similarity to at least one AI-generated reference review exceeds a threshold $\tau$, i.e., $\frac{1}{n} \sum_i \mathbb{I}[s_i > \tau]$; (ii) the mean of $s_{i,j}$ over all claim-reference pairs with $s_{i,j} > \tau$; (iii) the mean best-match similarity $\frac{1}{n} \sum_i s_i$; (iv) intra-review semantic diversity, defined as one minus the mean pairwise cosine similarity between claim embeddings within $\mathcal{C}_t$; and (v) the log-length of extracted claims: $\log(1 + |\mathcal{C}_t|)$.

Textual features capture token-level statistical properties of $t$, including perplexity, entropy, the proportion of tokens whose likelihood falls within the top-$k$ predictions of a language model, and the Fast-DetectGPT score.

Finally, we train a gradient-boosted decision trees classifier using the LightGBM framework (Ke et al., 2017). Hyperparameters are selected via randomized search with five-fold stratified cross-validation, optimizing macro-F1 to ensure balanced performance across the three classes.

# 4. Experiments

## 4.1. Implementation and Evaluation Setup

**Implementation.** We generate fully AI-written and LLM-refined reviews using four models: Gemini-2.5-Flash, Gemini-2.5-Pro, DeepSeek-V3.1, and Qwen3-235B-A22B (Comanici et al., 2025; Liu et al., 2024; Yang et al., 2025). Review cleaning and claim extraction is performed with Gemini-2.5-Flash; claim embeddings are obtained using Qwen3-0.6B (Zhang et al., 2025); and textual features are computed with Mistral-7B-Instruct-v0.3 as the reference model (Jiang et al., 2023). We use an 80%-20% train/test split, stratified by class and performed at the paper level,

ensuring that all reviews of a given paper appear exclusively in either the training or test set.

**Evaluation.** We evaluate Sem-Detect under two problem framings: binary classification, which distinguishes AI-generated reviews from non-AI ones, and three-class classification, which additionally separates LLM-refined human reviews as a distinct category. We report ROC curves, AUC, and True Positive Rates at 0.1% and 1% False Positive Rates for binary settings, and macro F1 for three-class. Where reported, uncertainty is estimated via bootstrap resampling (1,000 iterations).

## 4.2. Baselines

We compare Sem-Detect to general-purpose and domain-specific peer-review detectors.

On the general-purpose side, we evaluate LogRank (Ippolito et al., 2020), Fast-DetectGPT (Bao et al., 2024), Binoculars (Hans et al., 2024), MAGE (Li et al., 2024), and RADAR (Hu et al., 2023), spanning zero-shot, supervised, and adversarially-trained methods. Domain-specific baselines are the TF model (Kumar et al., 2024), Anchor (Yu et al., 2026) and EditLens (Thai et al., 2026). See Appendix C for details.

## 4.3. Research Questions

We evaluate Sem-Detect through experiments that address the following questions:

- **How competitive is Sem-Detect on the standard human vs. fully AI-generated task?** Since most prior works target binary authorship attribution, we first evaluate in a setting that excludes LLM-refined reviews (Section 5.1).

- **Can detectors flag fully AI-generated reviews without misclassifying legitimate LLM-assisted writing?** We study the three-class setting and quantify the trade-off between detecting fully AI reviews and avoiding false positives on AI-generated reviews (Section 5.2).

- **Can confidence-based filtering improve Sem-Detect's reliability in practice?** We analyze the accuracy/coverage trade-off when low-confidence predictions are flagged for manual review, rather than auto-classified (Section 5.3).

- **How robust is Sem-Detect to shifts in generation conditions?** We analyze out-of-distribution behavior under generation shifts by testing on both fully AI-generated and LLM-refined reviews produced by LLMs and prompting templates not used during training, and measure degradation relative to in-distribution evaluation (Section 5.4).

*Table 1.* Two-class detection (Human vs. AI). We report AUC and true positive rates (TPR) at fixed false positive rates (FPR) of 0.1% and 1%.

| Detector | AUC ↑ | TPR@0.1% ↑ | TPR@1% ↑ |
|---|---|---|---|
| LogRank | $0.576_{0.01}$ | $0.000_{0.00}$ | $0.001_{0.00}$ |
| MAGE | $0.699_{0.01}$ | $0.000_{0.00}$ | $0.008_{0.01}$ |
| Fast-DetectGPT | $0.699_{0.01}$ | $0.021_{0.01}$ | $0.062_{0.01}$ |
| Binoculars | $0.751_{0.01}$ | $0.008_{0.01}$ | $0.062_{0.02}$ |
| TF Model[†] | $0.926_{0.01}$ | $0.369_{0.06}$ | $0.553_{0.04}$ |
| RADAR | $0.965_{0.00}$ | $0.153_{0.05}$ | $0.371_{0.06}$ |
| Anchor[†] | $0.979_{0.00}$ | $0.541_{0.04}$ | $0.713_{0.07}$ |
| EditLens[†] | $0.998_{0.00}$ | $0.606_{0.16}$ | $0.956_{0.02}$ |
| Sem-Detect [†] | $\mathbf{0.999}_{0.00}$ | $\mathbf{0.760}_{0.11}$ | $\mathbf{0.973}_{0.03}$ |

[†] Domain-specific detectors trained or tuned on peer-review data.

*Figure 3.* ROC curves on the binary-setting. LLM-Refined reviews are not considered in this experiment.

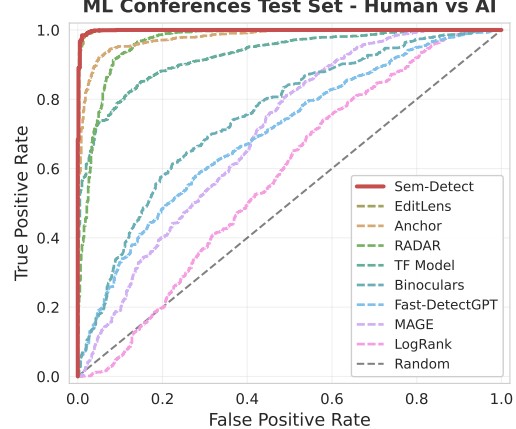

- **Does Sem-Detect generalize to a new peer-review domain without modification?** We apply Sem-Detect as-is to reviews from a medical imaging venue and measure cross-domain transfer relative to the standard ML-conferences test data (Section 5.5).

- **What does Sem-Detect predict on recent peer-review data?** We analyze authorship distributions on ICLR 2026 reviews, and compare trends with existing claims about AI prevalence in top-tier ML conferences (Section 5.6).

## 5. Results

### 5.1. Main Results: Binary Classification

Table 1 and Figure 3 summarize performance on the binary classification task, where LLM-refined reviews are not yet considered. In this setting, general-purpose detectors such as Binoculars and RADAR achieve moderate to strong AUC scores (0.751 and 0.965, respectively). However, their effectiveness declines at low false positive rates (FPR), which is critical for practical deployment. By contrast, domain-specific approaches are more robust in this region. The TF Model, Anchor and EditLens maintain competitive AUC while achieving higher true positive rates (TPR) at low FPR thresholds, underscoring the value of using signals specific to the peer-review domain.

Sem-Detect further improves on these results and performs best across all metrics. With an AUC of 0.999 and a TPR@0.1% FPR of 0.760 (a 25.5% relative improvement over EditLens), the results indicate that, even in the binary setting, combining claim-level semantic analysis with textual features improves performance over prior methods.

### 5.2. Main Results: Multi-Class Classification

The central contribution of our Sem-Detect lies in its ability to distinguish not only between human and AI authorship, but also to identify human reviews polished with an LLM.

**Comparison with binary detectors.** Most existing detectors produce only binary predictions. To compare against them, we first evaluate all methods under a simplified setting: we group LLM-refined and human reviews together as the non-AI class, while fully AI-generated reviews form the positive class. Figure 4 shows the results of this comparison.

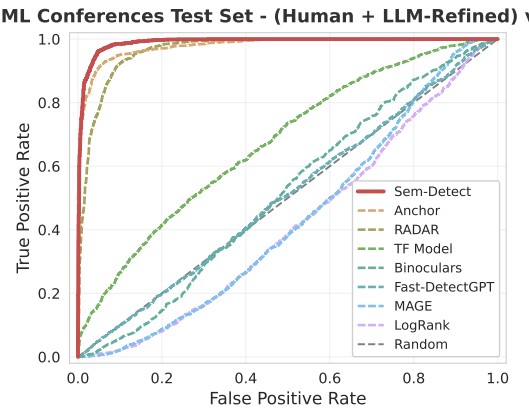

*Figure 4.* ROC curves for the collapsed binary task. Human and LLM-Refined reviews are grouped against fully AI reviews.

As shown in Figure 4, this setting proves challenging for most general-purpose detectors: LogRank, MAGE, Binoculars, and Fast-DetectGPT all collapse to near-random performance (AUC $\leq 0.513$). This outcome is expected: LLM-refined text shares many surface-level characteristics with fully AI-generated text, making it hard to separate the two classes based on textual features alone. The TF Model, de-

spite being tailored to the peer-review domain, also suffers a substantial drop (AUC = 0.674), as its reliance on token frequency patterns is disrupted by LLM refinement.

Two methods stand out as more robust. RADAR achieves an AUC of 0.966, suggesting that adversarial training helps the detector learn subtle differences between polished and fully generated text. Anchor also performs well (AUC = 0.980), which aligns with its emphasis on semantic similarity rather than surface-level patterns. However, neither method can distinguish among the three classes directly. Sem-Detect achieves the highest AUC (0.990) while also providing full three-class predictions.

**Three-class classification results.** We now turn to the main evaluation setting. Figure 5 reports the confusion matrix for Sem-Detect on the three-class task.

Overall, the classifier performs well on both AI-generated and LLM-refined reviews, correctly identifying 91.18% of AI reviews and 91.61% of LLM-refined ones.

The main source of error involves human-written reviews being classified as LLM-refined (35.38%), likely reflecting both the inherent difficulty of separating polished human writing from LLM-assisted text and the class imbalance in training, where LLM-refined reviews are more prevalent.

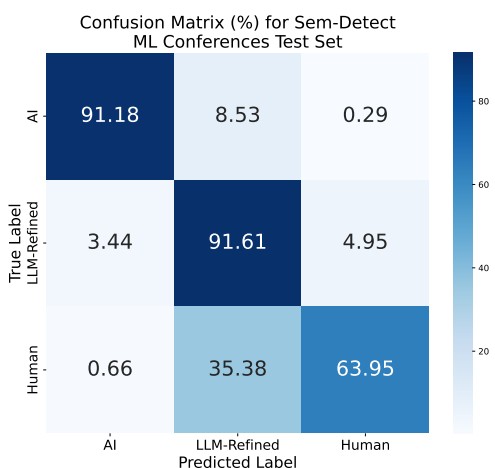

Figure 5. Sem-Detect Multi-Class Confusion matrix (%).

We view this error pattern as acceptable because the resulting bias is conservative: when uncertain, the model tends to predict LLM-refined rather than fully AI-generated. As a result, hard misclassifications from human to AI remain very rare (0.66%), which is desirable in practice.

**The role of semantic similarity.** Figure 6 illustrates why claim-level analysis proves effective. The plot displays the mean best-match claim similarity for each class (the most discriminative feature in our classifier). AI-generated reviews show consistently high similarity to reference AI

reviews (median $\approx 0.73$). Human and LLM-refined reviews, by contrast, cluster together at lower values (median $\approx 0.64$), hence supporting our premise: AI models converge on similar claims, but LLM refinement preserves the distinctiveness of human judgments.

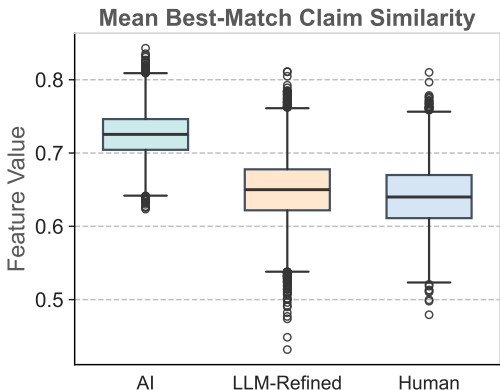

Figure 6. Mean best-match claim similarity by class (test set).

### 5.3. Deployment via Confidence Thresholding

By default, Sem-Detect predicts the highest-probability class regardless of certainty. For example, probabilities of 0.51 AI-generated, 0.48 human-written, and 0.01 LLM-refined still yield an AI-generated label, despite near-tie uncertainty between human and AI, which is undesirable when false accusations are costlier than missed detections.

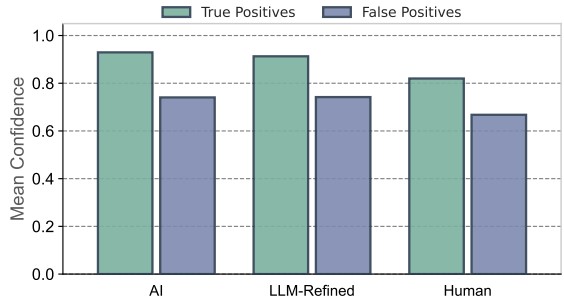

Figure 7. Prediction confidence calibration by predicted class.

Fortunately, as Figure 7 shows, Sem-Detect's confidence scores are well-calibrated: correct predictions average 0.91 confidence while incorrect ones average 0.72.

We can therefore introduce a confidence threshold $\theta$ that flags low-confidence predictions for manual review, trading coverage for accuracy on the rest.

Figure 8 further quantifies the trade-off. At $\theta = 0.80$, 79% of reviews are still classified automatically and accuracy on that set rises to 94.7%, while the Human → LLM-refined error, the main failure mode in Figure 5, drops substantially.

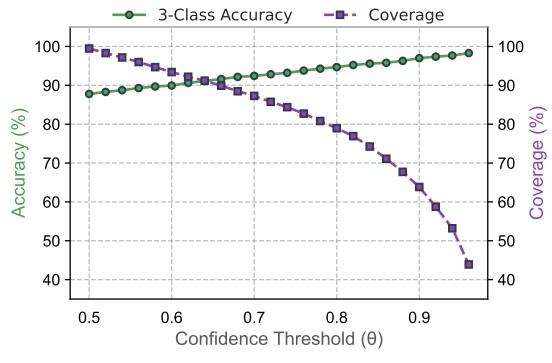

*(a)* Accuracy-coverage trade-off.

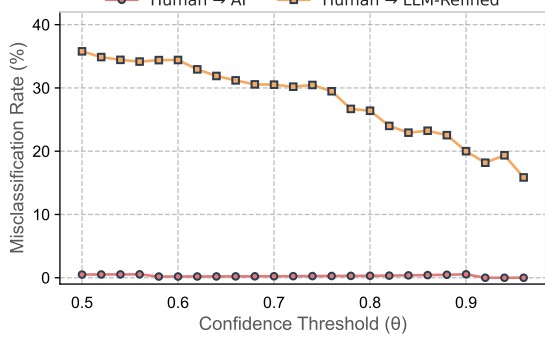

*(b)* Human review misclassification rates.

*Figure 8.* Effect of confidence thresholding on classification accuracy, coverage, and Human → LLM-refined error rate.

## 5.4. Robustness to Generation Conditions

In practice, reviewers may use diverse models and prompts to generate or refine reviews, raising the question of whether Sem-Detect generalizes beyond its training conditions. We evaluate two out-of-distribution settings: (i) OOD-M, where reviews are generated by unseen model families using the same prompt template, and (ii) OOD-M+P, where both models and prompts differ. For OOD-M, we use Mistral-Large-3 (Mistral, 2025), Claude-Sonnet-4 (Anthropic, 2025), and GPT-oss-120b (Agarwal et al., 2025); for OOD-M+P, we additionally vary prompt structure, specificity, and review format (Further details in Appendix D.1). Table 2 reports three-class performance under these conditions.

*Table 2.* Sem-Detect under distribution shift, for two settings: (i) different models (M) and (ii) different models and prompts (M+P).

|  | In-Dist | OOD-M | OOD-M+P |
|---|---|---|---|
| Same Models as Train | ✓ | ✗ | ✗ |
| Same Prompts as Train | ✓ | ✓ | ✗ |
| AI Precision | 0.93 | 0.97 | 0.96 |
| AI Recall | 0.91 | 0.67 | 0.65 |
| 3-Class Macro-F1 Score | 0.84 | 0.71 | 0.68 |

### 5.4.1. PERFORMANCE UNDER DISTRIBUTION SHIFT

We expected performance to drop under distribution shift, and it does: 3-class Macro-F1 falls from 0.84 to 0.71 (OOD-M) and 0.68 (OOD-M+P). What matters, though, is how the model fails. Rather than making high-stakes errors, Sem-Detect routes uncertain samples to the LLM-refined class, and overall, AI precision actually increases to 0.97, meaning predictions of "AI-generated" are highly reliable.

This conservative behavior raises a natural question: is LLM-refined merely an uncertainty bucket? The OOD class-wise metrics suggest otherwise. In fact, under OOD-M+P, the LLM-refined class achieves a recall of 0.769 and a precision of 0.759, a pattern inconsistent with a catch-all category, which would typically show degradation in at least one of these metrics (further details in Appendices D.2-D.5).

## 5.5. Cross-Domain Generalization

We now extend our evaluation to a different field: medical imaging. We select MIDL 2022 for this analysis because, like ICLR and NeurIPS, it hosts its reviews on OpenReview, allowing us to collect authentic human reviews under the same conditions. Specifically, we sample ≈ 100 random papers from this venue, generate AI-written and LLM-refined reviews using our standard pipeline, and run Sem-Detect without any modifications.

The results are very positive. As Figure 9 shows, Sem-Detect achieves comparable or slightly higher F1 scores on MIDL than on the ML conferences test set. This holds across all three classes. That said, one limitation deserves mention: MIDL, while medically oriented, still centers on deep learning methods. Evaluating on more distant fields would be ideal, but open peer-review data remains limited outside of computer science.

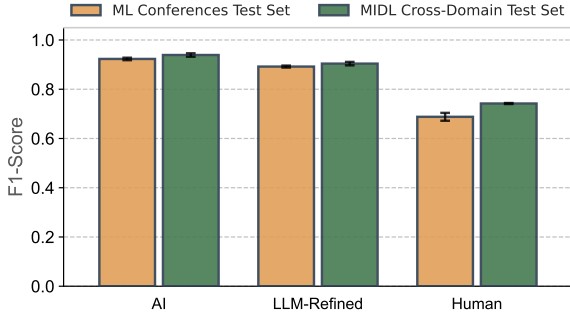

*Figure 9.* Cross-domain generalization results. F1 scores for Sem-Detect on the ML test set and the medical imaging venue MIDL 2022. No domain-specific retraining is performed.

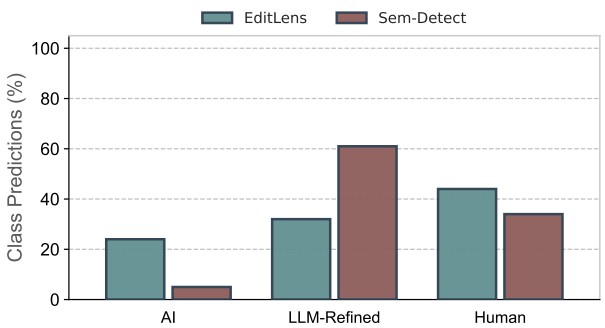

*Figure 10.* Sem-Detect and EditLens (Pangram Labs) review authorship predictions on ICLR 2026 data.

### 5.6. ICLR 2026 Comparison

Our evaluations so far have relied on data where ground truth labels are known due to temporal constraints. To examine how Sem-Detect behaves in a contemporary setting, we turn to ICLR 2026, sampling approximately 600 papers at random. This analysis is motivated by recent claims from Pangram Labs (Thai et al., 2026), whose EditLens detector suggests that more than 20% of ICLR 2026 reviews were fully AI-generated (Emi, 2025). Without ground truth, our goal is not to establish which method is correct. Instead, we examine whether Sem-Detect produces a reasonable distribution of review categories.

Figure 10 shows that the two methods present quite different distributions: EditLens classifies 24% of reviews as AI-generated, 32% as LLM-refined, and 44% as human; Sem-Detect predicts 5%, 61%, and 34%, respectively. The divergence appears primarily in how each method handles the middle ground: while EditLens places predictions more liberally on the extreme classes, Sem-Detect favors LLM-refined classifications for ambiguous cases. This conservative behavior ends up being desirable in practice as, in high-stakes settings, false accusations carry greater cost than missed detections.

That said, both distributions appear plausible, and for reviews that Sem-Detect classifies as either fully AI-generated or fully human, EditLens agrees with the prediction approximately 70% of the time. This suggests that, despite their different design philosophies, both methods capture meaningful signal about AI presence in peer review.

### 6. Conclusions

In this paper, we propose Sem-Detect, a detection framework for peer-review authorship attribution that distinguishes fully human-written reviews from those refined using an LLM and those generated end-to-end by a machine. Our approach exploits the fact that authorship signals reside not only in textual features of the review, but also in the semantic content of expressed ideas. While different AI models tend to converge on similar claims when reviewing the same paper, human reviewers introduce more unique and diverse judgments.

We validate Sem-Detect on reviews from top-tier ML conferences and find that it outperforms all baselines in both binary and three-class settings. At the same time, fewer than 3.5% of LLM-refined human reviews are mistakenly flagged as AI-generated.

Beyond these controlled conditions, Sem-Detect also shows reasonable behavior under distribution shift. The method generalizes to unseen models, transfers to medical imaging reviews without retraining, and produces plausible predictions on recent ICLR 2026 data. This shows that effective detection and fairness to legitimate LLM use can coexist.

### Impact Statement

This work contributes to the ongoing effort to preserve integrity in peer review. By distinguishing fully AI-generated reviews from those where humans used an LLM only to improve clarity, our framework supports policies that can detect problematic content without penalizing responsible AI assistance.

That said, we recognize an important limitation in our approach. Our method assumes that the originality of ideas can help distinguish human from AI authorship. As models continue to improve, they may eventually produce reviews with novel, high-quality insights that are indistinguishable from, or even better than, those of human experts. If that happens, the line between human and AI authorship may blur, raising a deeper question: does the origin of an idea matter if its quality is sound?

Finally, we note that any detection system risks false accusations, which can harm reviewers' reputations. While our results show very low rates of misclassification between true human entries and AI, we emphasize that our method should be used as one signal among many, not as a definitive judgment.

### Reproducibility

We release the following artifacts:

- 🔗 **Code**: full pipeline, two pre-trained classifiers, and a self-hosted Flask web demo.

- 🤗 **Data**: complete set of reviews for the 800 papers from ICLR and NeurIPS 2021-2022.

Further details on prompts, data construction, and additional analyses are provided in the appendices.

## Acknowledgements

We acknowledge support from national funds through Fundação para a Ciência e a Tecnologia, I.P. (FCT), under projects UID/50021/2025 and UID/PRR/50021/2025.

This work is also co-financed by FCT through the Carnegie Mellon Portugal Program under the fellowship PRT/BD/155049/2024.

Lei Li is partly supported by the CMU CyLab seed grant.

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

# A. Dataset Creation Details

## A.1. Data Statistics

Table 3 summarizes the scale and composition of our dataset across venues and years. We see that the average number of extracted claims per review is stable between human and LLM-refined reviews, indicating that refinement preserves the underlying semantic structure, and we see fully AI-generated reviews consistently containing more claims per review, reflecting their tendency to produce longer, more exhaustive feedback. Figure 11 complements these statistics by showing that claim type distributions are largely consistent across conferences and years, with evaluation and constructive input forming the majority of content.

*Table 3.* Dataset statistics by review class.

| Dataset | Reviews | | | Claims | | | Avg Claims/Review | | |
|---|---|---|---|---|---|---|---|---|---|
| | Human | Rewrite | AI | Human | Rewrite | AI | Human | Rewrite | AI |
| ICLR 2021 | 782 | 3,128 | 1,788 | 18,142 | 73,167 | 58,083 | 23.20 | 23.39 | 32.48 |
| ICLR 2022 | 773 | 3,092 | 1,456 | 19,205 | 76,531 | 47,630 | 24.84 | 24.75 | 32.71 |
| NeurIPS 2021 | 778 | 3,184 | 1,780 | 17,517 | 71,150 | 56,622 | 22.52 | 22.35 | 31.81 |
| NeurIPS 2022 | 732 | 2,928 | 1,744 | 15,640 | 62,152 | 56,284 | 21.37 | 21.23 | 32.27 |
| **Total** | **3,065** | **12,332** | **6,768** | **70,504** | **283,000** | **218,619** | **22.98** | **22.93** | **32.34** |

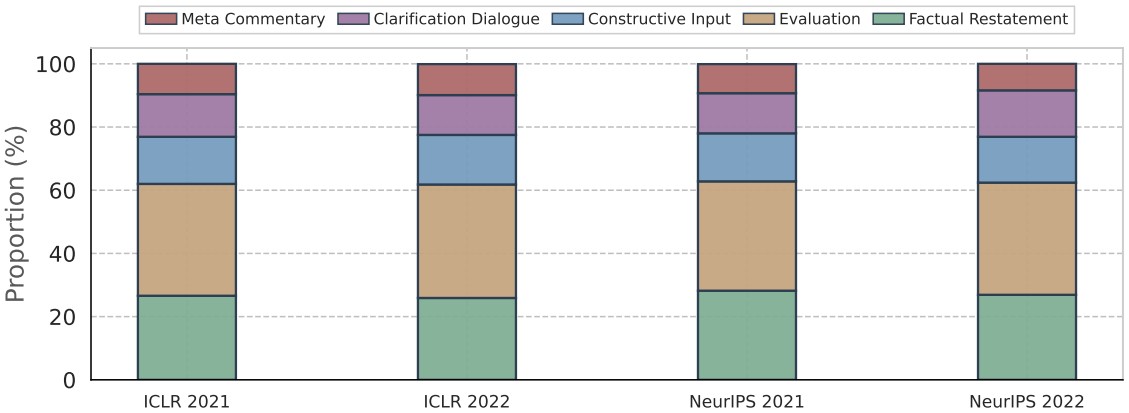

*Figure 11.* Distribution of claim types across venues and years.

## A.2. Generating Fully AI-Reviews

Table 4 presents the prompt template used to generate AI-Reviews. We ensure a maximum output length of 3,072 tokens and a temperature of 1.0. The goal is to encourage diversity in the generated reviews while still producing coherent evaluations.

*Table 4.* System Prompt used to Generate the AI Reviews.

---

**Generating AI Reviews - System Prompt**

Review the given paper for a top AI conference.

Please be concise, critical, focused, and constructive so that the authors find the review convincing and improve their manuscript accordingly.

Your final recommendation should be "{score}". Please write a review that includes:

(1) Summary of the paper; (2) Strengths; (3) Weaknesses; (4) Questions for authors (if any) and (5) Final Judgement.

---

## A.3. Generating LLM-Refined Reviews

Table 5 presents the prompt template used for this task. Similarly to the fully AI-generated reviews we use a maximum output length of 3,072 tokens but, we use a temperature of 0.8 instead. The slightly lower temperature (compared to 1.0 for fully AI-generated reviews) is to encourage the model to stay closer to the source text while still allowing stylistic variation.

*Table 5.* System Prompt used to Generate the LLM-Refined Reviews.

---

### Generating LLM-Refined Reviews - System Prompt

You are a professional writing assistant.

Your task is to take user-provided text and rewrite it to be more polished, professional, and effective.

Ensure the tone is appropriate for academic communication.

**Do not modify the content of the review or suggest any improvements.**

---

## A.4. Extracting Claims from Reviews

As described in Section 3.1, we extract structured claim-level representations from each review using Gemini-2.5 Flash. Table 6 presents the prompt template used to perform the extraction.

*Table 6.* System Prompt used to Extract Claims from Reviews.

---

### Extracting Claims from Reviews - System Prompt

You are given the full text of a peer review.

Your task is to extract and organize the reviewer's comments into bullet points under the following categories:

1. **Factual Restatement** – Summaries or descriptions of what the paper does, its methods, contributions, or results.

2. **Evaluation** – Judgments of quality, including both strengths (positive evaluations) and weaknesses/limitations (negative evaluations).

3. **Constructive Input** – Actionable suggestions or recommendations for improvement.

4. **Clarification Dialogue** – Questions directed to the authors or requests for clarification.

5. **Meta-Commentary** – Remarks about the broader context, such as fit for the venue, clarity of writing, novelty, or overall recommendation.

---

On Tables 7 and 8 we now illustrate the claim extraction process with a real example: the full original review text is shown first, followed by its extracted claims, with color coding to highlight the correspondence between source passages and their derived claims.

*Table 7.* Complete example of an original human-written peer review from ICLR 2021.

---

### Original Review - Fully Human

This paper analyses the random shooting control strategy in combination with various generative models, which model observations conditioned on the history of observation-action pairs.

The authors select two variants of the Acrobot environment to make requirements like multimodal posteriors more explicit. Both statistics on the distribution associated with the generative model under a fixed policy and reward-dependent metrics were defined and analysed for a range of models.

The paper's contribution are twofold. First, the suggested experimental protocol for model evaluation and benchmarking extends the usual evaluation process in reinforcement learning, which often focuses purely on the cumulative reward. The framework described introduces a range of static"likelihood-based metrics. These static metrics are evaluated under a fixed (potentially stochastic) policy and allow the separation of the model-based control strategy and the underlying model for evaluation purposes. Reward-based dynamic" metrics are evaluated under a random-shooting control mechanism. This framework heavily simplifies model evaluation by providing evaluation metrics and fixing the environment and control strategy. Under these assumptions this approach allows direct comparison and even visualisation of various quantities of interest, including trajectories of the one-step forward model. Second, this work applies the conceived framework to evaluate a range of models on two different environments. These environments are intended to make requirements like probabilistic posteriors explicit. The authors conclude by claiming that 1. Probabilistic models are needed when the system benefits from multimodal predictive uncertainty", and 2. Deterministic models are sufficient if trained with a loss allowing heteroscedasticity".

There's an inherent trade-off between simplicity of the study and generality its conclusion. While some of the simplifying assumptions made make this kind of study possible in the first place, they also raise a range of questions:

- How appropriate are these metrics for problems with higher observation spaces? Can we expect the variance of estimates to increase and ratios and likelihoods to diminish by multiple orders of magnitude?

- Do the claims presented as important findings" generalise to other environments?

- Does the correlation of the explained variance with the dynamic metrics hold on other environments?

A clear definition of micro-data reinforcement learning is missing, and MBRL is introduced twice with conflicting definitions.

The future directions outlined by the authors of extending the results to larger systems and other planning strategies are very relevant to reduce the concerns of generalisability of the results and applications to problems of higher dimensions. At the same time these modification will increase the experimental complexity. This paper serves as a suitable baseline for reference for future work to answer these questions. Therefore I consider this paper a valid contribution to ICLR.

*Table 8.* The same review after the claim extraction process. For readability, some factual-restatement claims are omitted for space.

| Claim Extraction Example of Fully Human Review |
| --- |

**Factual Restatement**

- This paper analyses the random shooting control strategy in combination with various generative models which model observations conditioned on the history of observation-action pairs.

- The authors select two variants of the Acrobot environment to make requirements like multimodal posteriors more explicit.

- Both statistics on the distribution associated with the generative model under a fixed policy and reward-dependent metrics were defined and analysed for a range of models.

- The paper's contribution is twofold: first, the suggested experimental protocol for model evaluation and benchmarking extends the usual evaluation process in reinforcement learning, which often focuses purely on the cumulative reward.

- The framework described introduces a range of "static" likelihood-based metrics, which are evaluated under a fixed policy and allow the separation of the model-based control strategy and the underlying model for evaluation .

- Reward-based dynamic metrics are evaluated under a random-shooting control mechanism.

- Second, this work applies the conceived framework to evaluate a range of models on two different environments, which are intended to make requirements like probabilistic posteriors explicit.

- The authors conclude by claiming that 1. Probabilistic models are needed when the system benefits from multimodal predictive uncertainty, and 2. Deterministic models are sufficient if trained with a loss allowing heteroscedasticity.

**Evaluation**

- There's an inherent trade-off between simplicity of the study and generality its conclusion.
- While some of the simplifying assumptions made make this kind of study possible, they also raise a range of questions.
- The future directions outlined by the authors of extending the results to larger systems and other planning strategies are very relevant to reduce the concerns of generalisability of the results and applications to problems of higher dimensions.
- This paper serves as a suitable baseline for reference for future work to answer these questions.

**Constructive Input**

- A clear definition of micro-data reinforcement learning is missing.
- MBRL is introduced twice with conflicting definitions.

**Clarification Dialogue**

- How appropriate are these metrics for problems with higher observation spaces?
- Can we expect the variance of estimates to increase and ratios and likelihoods to diminish by multiple orders of magnitude in higher observation spaces?
- Do the claims presented as "important findings" generalise to other environments?
- Does the correlation of the explained variance with the dynamic metrics hold on other environments?

**Meta Commentary**

- I consider this paper a valid contribution to ICLR.

### A.5. Cost Analysis: Review Generation, Cleaning and Claim Extraction

Figure 12 summarizes the computational costs for review generation and cleaning. Whenever possible, we use batch API calls to reduce latency and cost. In our case, Gemini models are queried with Gemini Batch API requests[2], while DeepSeek and Qwen-3 use synchronous requests through AWS[3]. The generation step (Figure 12a) represents the largest expense, with total costs approaching $170. These costs, however, are distributed roughly evenly between fully AI-generated and LLM-refined reviews, despite the latter being far more numerous. This happens because AI reviews require the parsed PDF as input, while LLM-refined reviews only receive the shorter human review. The cleaning stage (Figure 12b) results in lower costs, as it involves only rewriting reviews to remove formatting artifacts that would otherwise reveal their LLM origin.

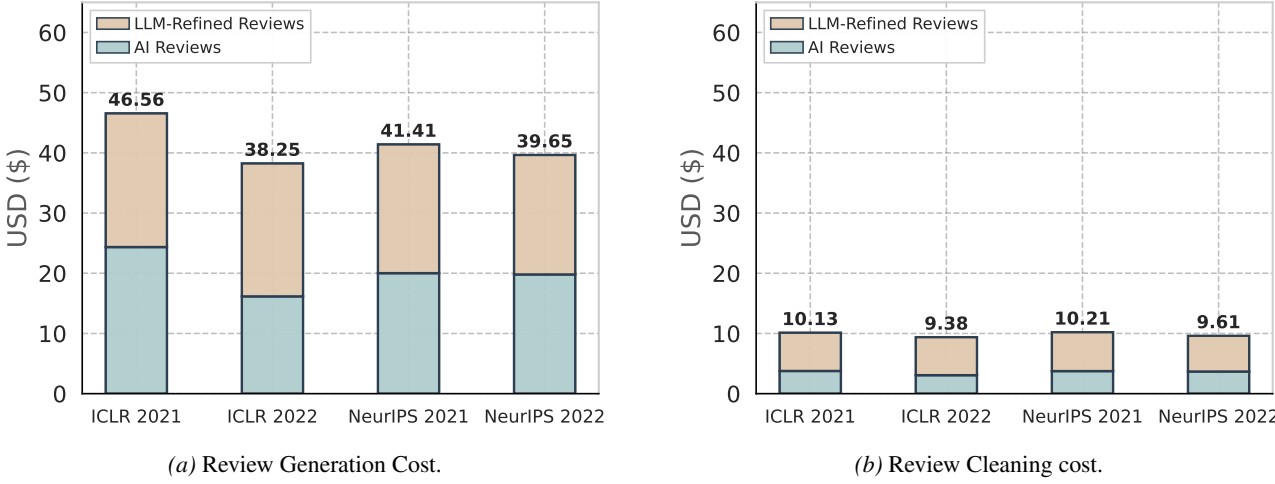

*(a) Review Generation Cost.*   *(b) Review Cleaning cost.*

*Figure 12.* Computational costs for (a) review generation and (b) review cleaning, broken down by venue and year.

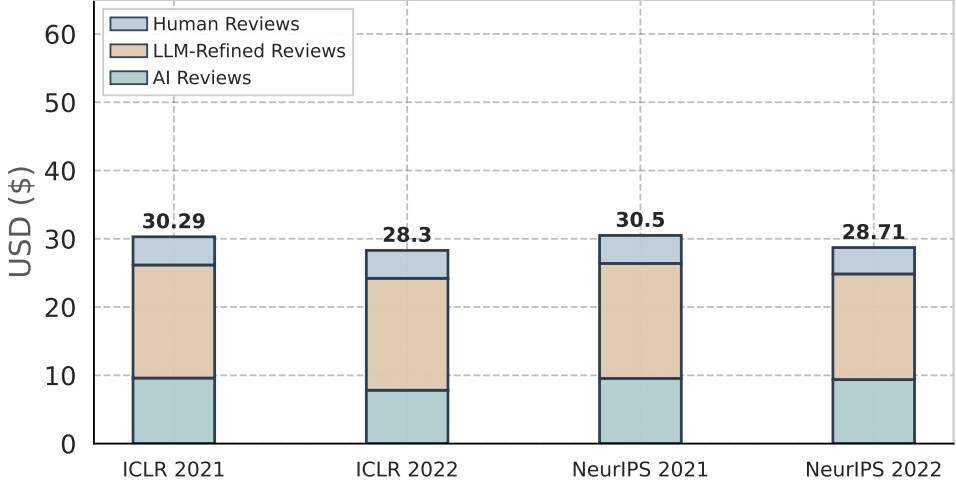

*Figure 13.* Computational cost for claim extraction across review classes, broken down by venue and year.

Figure 13 reports claim extraction costs across all three review classes. As expected, LLM-refined reviews dominate expenses due to their larger volume in our dataset (four refinements per human review). With that said, this cost structure applies only to classifier training. In a future deployment setting, claim extraction would only be needed for incoming reviews and AI-generated references, which would reduce inference-time overhead.

---

[2]https://ai.google.dev/gemini-api/docs/batch-api
[3]https://docs.aws.amazon.com/bedrock/latest/userguide/models-supported.html

# B. Design Choice Ablations

## B.1. Selecting the Right Classifier

To combine our nine features into final predictions, we compared three classifiers: XGBoost, LightGBM, and Random Forest. For each one, we performed randomized hyperparameter search with five-fold stratified cross-validation, using macro-F1 as the optimization target. Figure 14 shows the resulting test-set performance across all configurations. As the boxplots indicate, median scores are similar across the three models, all falling between 0.83 and 0.84. However, the models differ in how sensitive they are to hyperparameter choices. Random Forest, in particular, produces several outliers below 0.79, while XGBoost and LightGBM remain more stable.

Based on these results, we selected LightGBM for our final model. Although its median performance is only slightly higher than that of XGBoost, its outliers stay closer to the central distribution, suggesting more consistent behavior regardless of the specific hyperparameter configuration. Table 9 lists the hyperparameters of the best-performing model.

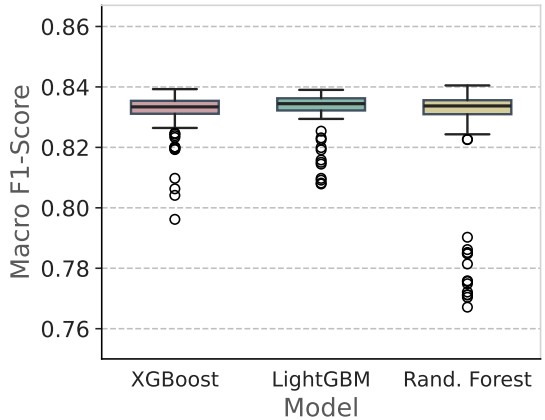

*Figure 14.* Comparison of classifier performance across hyperparameter configurations. Each boxplot shows the distribution of macro-F1 scores on the test set obtained during randomized search.

*Table 9.* LightGBM hyperparameters for the best-performing model.

| Parameter | Value |
|---|---|
| Boosting type | Gradient Boosted Decision Trees |
| Objective | Multiclass classification |
| Number of classes | 3 |
| Number of trees | 100 |
| Learning rate | 0.1 |
| Maximum tree depth | 7 |
| Number of leaves | 15 |
| Subsample ratio | 0.6 |
| Feature subsample ratio | 1.0 |
| Minimum samples per leaf | 30 |
| Minimum split gain | 0.2 |
| L1 regularization | 1.0 |
| L2 regularization | 1.0 |

## B.2. Number of Reference Reviews

Sem-Detect, by default, pairs each target review with $k = 3$ AI-generated reference reviews of the same paper. In this section we ablate whether three references are necessary, or whether fewer would be enough. As such, we train and evaluate Sem-Detect with $k$ ranging from 1 to 3, keeping all other settings fixed.

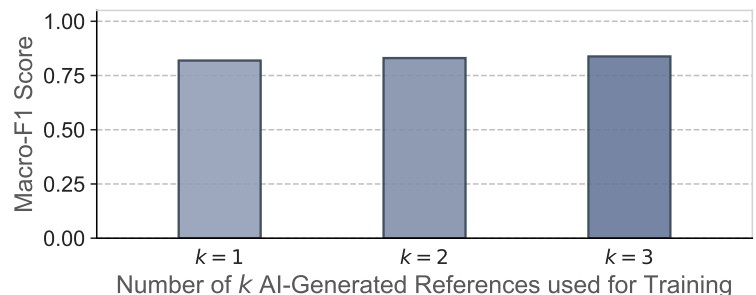

*Figure 15.* Effect of the number of reference reviews ($k$) on three-class detection performance.

From Figure 15 we observe that performance improves monotonically with $k$, but even a single reference review achieves a Macro-F1 of 0.819. We use $k = 3$ in our main experiments as it offers the best performance, but users seeking lower inference cost or latency could train with smaller values of $k$ knowing this trade-off.

## B.3. Embedding Model Choice

In this section, we describe two sets of experiments that guided our choice of embedding model. First, we examine how model size affects performance within a single model family. Second, we compare different embedding model families to assess whether our choice generalizes across architectures.

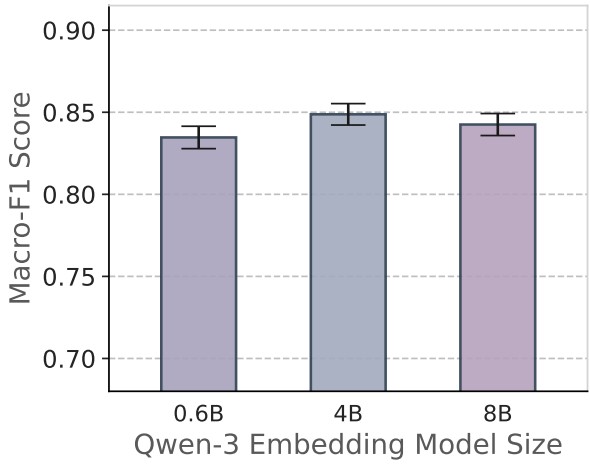 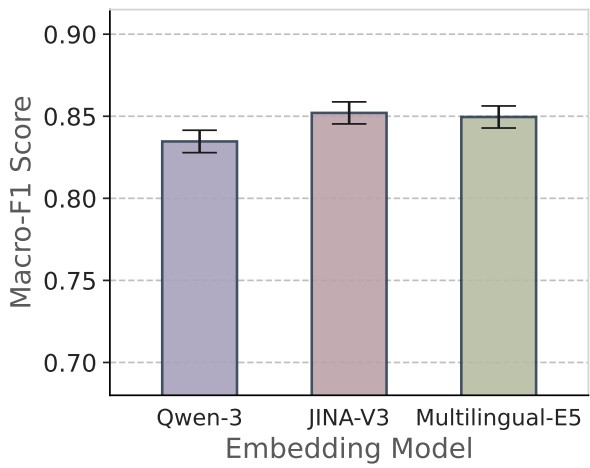

*(a)* Effect of model size within the Qwen-3 family.      *(b)* Comparison across embedding model families.

*Figure 16.* Analysis of embedding model choice on the three-class detection performance.

### B.3.1. SCALING EMBEDDING MODEL SIZE

We evaluate three variants of Qwen-3 Embedding (Zhang et al., 2025) at different scales: 0.6B, 4B, and 8B parameters. Figure 16a presents the results.

While performance improves as model size increases, the benefits plateau at the 4B scale, as the 8B model performs on par with the 4B variant. Despite these findings, we report our main experiments using the 0.6B model.

This decision reflects practical considerations: the smaller model is substantially faster to run and requires less storage, making it more accessible for reproducibility.

### B.3.2. TESTING DIFFERENT EMBEDDING MODELS

Beyond model size, we also investigate whether our method is sensitive to the choice of the embedding model family. To this end, we compare three top-performing models of similar size according to the MTEB Leaderboard[4]: Qwen-3 Embedding, JINA-V3, and Multilingual-E5, all at approximately 0.6B parameters (Yang et al., 2025; Sturua et al., 2025; Wang et al., 2024). As shown in Figure 16b, all three models perform similarly, with Macro-F1 scores ranging from 0.84 to 0.85. JINA-V3 and Multilingual-E5 achieve, nevertheless, marginally higher scores than Qwen-3.

Given that we had already conducted multiple experiments before running this comparison, and since the performance gap is minimal, we present our main results using Qwen-3. This comparison, however, demonstrates that Sem-Detect generalizes well across embedding architectures, giving users the freedom to select models that best fit their preferences. One important detail: since we retrain the classifier from scratch for each embedding model, users who wish to use Sem-Detect with an alternative architecture should expect to repeat the training step.

---

[4]https://huggingface.co/spaces/mteb/leaderboard

### B.4. Claim Extraction vs Raw Review Text with Sentence-Level Chunks

A main design choice in Sem-Detect is the use of LLM-based claim extraction from the reviews. This approach, however, results in further computational costs, as each review must be processed an additional time by another LLM. A natural question is whether this step is truly necessary, or whether other segmentation strategies could achieve better results.

In Section 5, we show that not segmenting at all, as Anchor (Yu et al., 2026) does when embeds entire reviews, performs quite well, but not as good as Sem-Detect. Here, we explore the opposite direction: segmenting at a finer granularity using sentence-level chunking, which splits reviews at default sentence boundaries and produces chunks of more comparable length to our LLM-extracted claims. As shown in Figure 17, the two approaches perform similarly on human reviews. However, the gap becomes substantial for the other two classes. In particular, for fully AI-generated reviews, claim-level segmentation proves far more effective than its sentence-level counterpart.

We believe that this gap happens because sentence boundaries do not always align with semantic boundaries. Table 10 illustrates one such failure case: over-segmentation. In this example, two consecutive sentences from a single argument get split by sentence-level chunking, which breaks their semantic relation. Our claim-level approach, by contrast, recognizes they belong together and groups them as one unit. When claims are fragmented in this way, similarity comparisons become noisier, and consequently reduce the method's performance.

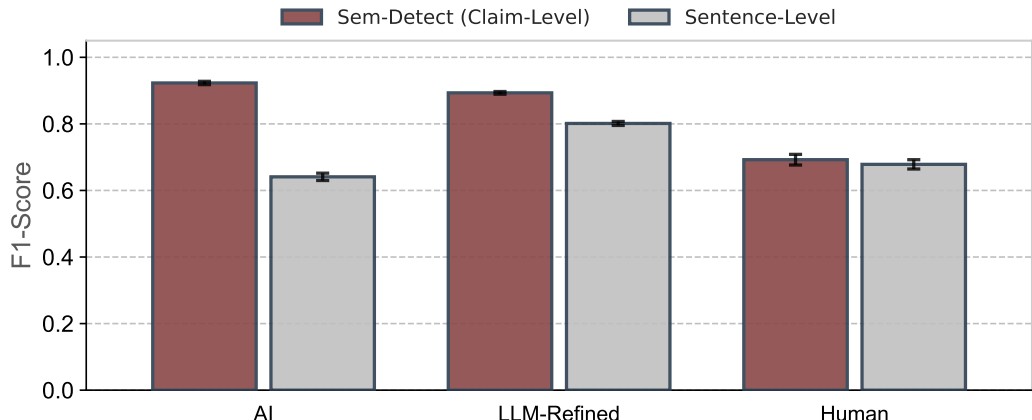

*Figure 17.* F1 score comparison between claim-level and sentence-level segmentation strategies.

*Table 10.* Sem-Detect understands both sentences belong to the same idea and groups them together.

---

### Example of Failure Case for Sentence-Level Chunks: Over-segmentation

**Review Verbatim**

Non-training data is limited to 5 books released in 2025, with no diversity in genre or timeframes. This makes it hard to validate the method's robustness against false positives across non-training scenarios.

---

**Sentence-Level Chunking**

• Non-training data is limited to 5 books released in 2025, with no diversity in genre or timeframes.

• This makes it hard to validate the method's robustness against false positives across non-training scenarios.

**Sem-Detect Claim-Level Segmentation**

Non-training data is limited to 5 books released in 2025, with no diversity in genre or timeframes, making it hard to validate RECAP's robustness against false positives across non-training scenarios.

## B.5. Feature Selection and Interpretability

### B.5.1. Feature Importance and Distribution

As introduced in Section 3.2, our classifier is based on a total of nine discriminative features. Figure 18 reports the feature importance scores assigned by LightGBM. While the main paper provides the formal definitions of these features, this section offers additional intuition for their inclusion. We first discuss each feature, and then analyze the empirical distributions of the most discriminative ones using box plots.

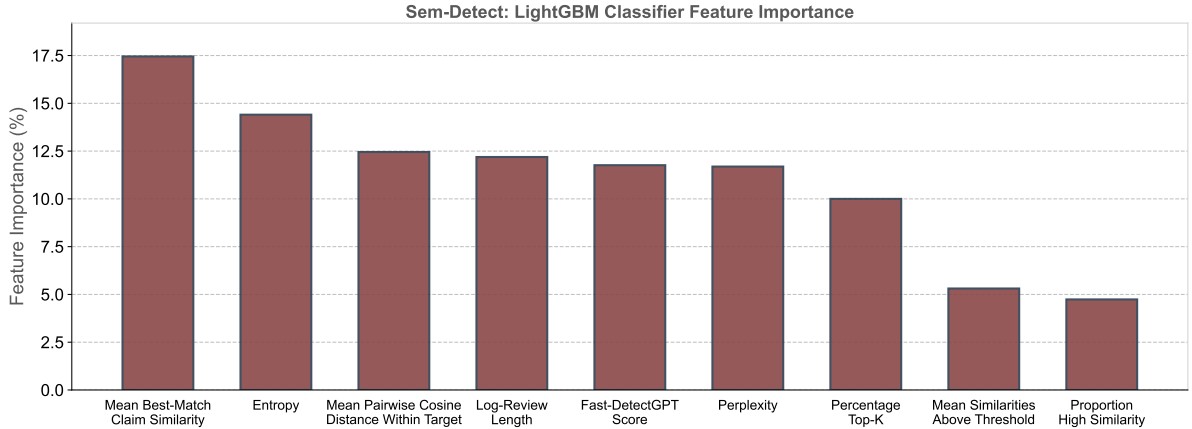

*Figure 18.* Relative importance of the nine features as learned by the LightGBM classifier.

1. **Proportion of High-Similarity**: Captures what proportion of a review is highly aligned with the AI-generated references. For each target claim, we check whether its maximum semantic similarity to any claim in the AI-generated references exceeds a threshold $\tau$, and report the fraction of claims that do so. We tune $\tau$ via a linear sweep from 0.7 to 0.9 during training, and fix $\tau = 0.8$ for all reported results.

2. **Mean Similarities Above Threshold**: For the subset of claims previously identified as having strong overlap with the AI-generated references, this feature captures how strong that overlap is on average. For all target claims whose maximum semantic similarity to any AI claim exceeds $\tau$, we compute the mean of these maximum similarity values.

3. **Mean Best-Match Claim Similarity**: Captures the overall semantic proximity of a review to AI-generated content. For each target claim, we compute its best-match semantic similarity to any claim in the AI-generated reference reviews, and then average these best-match similarities across all target claims.

4. **Intra-Review Semantic Diversity**: Captures how semantically varied the claims within a review are. We compute all pairwise cosine similarities between claim embeddings within the target review and define the feature as one minus their mean, so that higher values correspond to greater semantic diversity and lower redundancy.

5. **Log Review Length**: Captures the effective length of a review while reducing the influence of very long outliers. We compute the natural logarithm of one plus the number of claims extracted from the target review.

6. **Entropy**: Captures uncertainty in the language model's next-token predictions along the review. We average the entropy of the model's next-token distribution over all positions in the text.

7. **Perplexity**: Captures how predictable the review text is under a given language model. Although entropy and perplexity are closely related, we include both features as they capture complementary aspects of the model's behavior, and we find that including both consistently improves classification performance in practice.

8. **Top-$k$ Token Percentage**: Captures how often the review follows highly probable token choices under a language model. We compute the fraction of tokens in the target review whose next-token probability lies within the model's top-$k$ predictions, using $k = 200$.

9. **Fast-Detect Score**: Captures token-level statistical signals associated with machine-generated text. As an additional textual feature, we include the score produced by Fast-DetectGPT when applied to the target review.

While Figure 18 reveals which features the classifier relies on most, it does not explain why these features are discriminative. To address this, Figure 19 presents the distributions of the four most important features across the three classes.

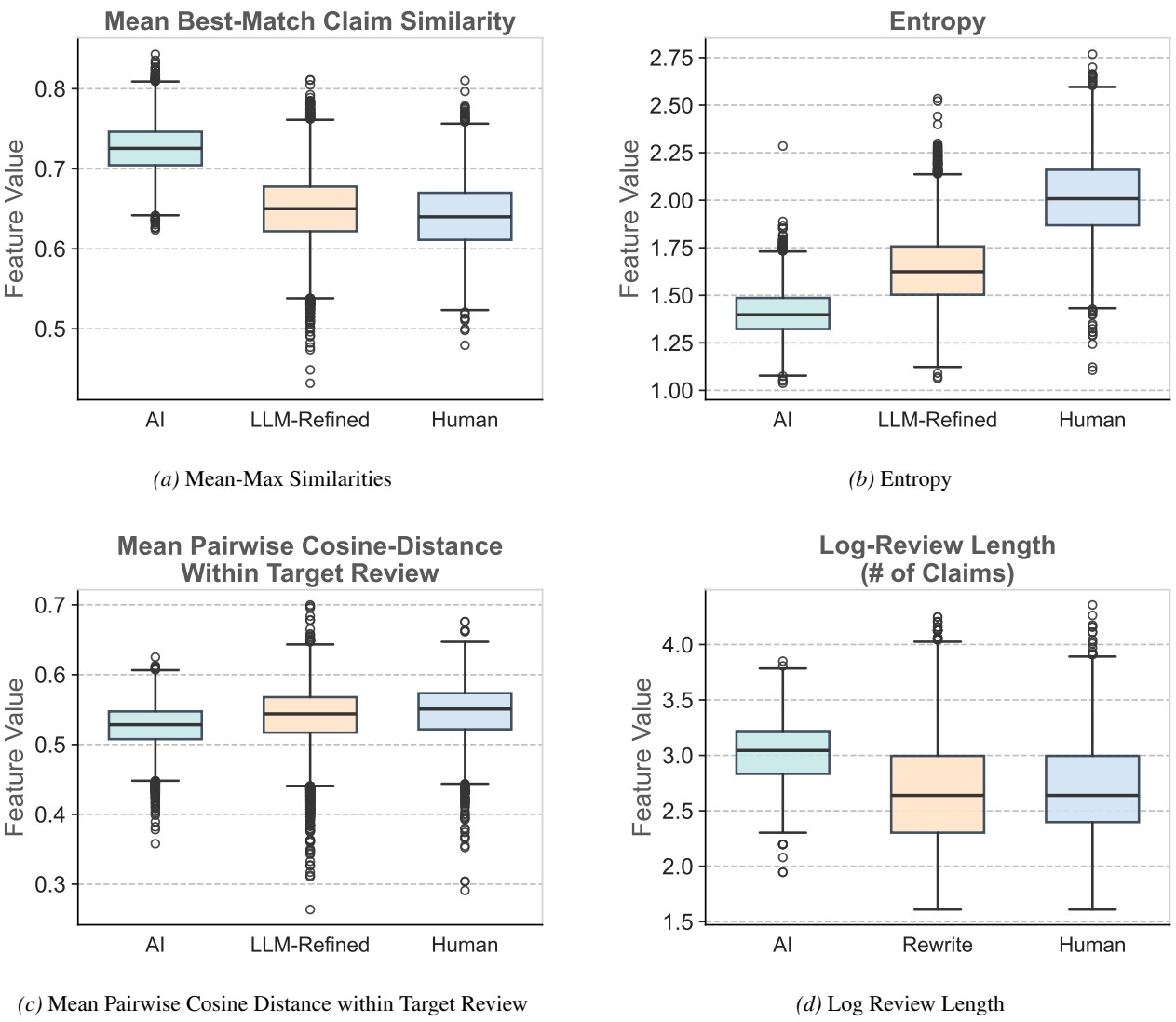

*(a)* Mean-Max Similarities

*(b)* Entropy

*(c)* Mean Pairwise Cosine Distance within Target Review

*(d)* Log Review Length

*Figure 19.* Distribution of semantic and surface-level features across review types.

A clear pattern emerges from these distributions. Semantic features primarily separate fully AI-generated reviews from the other two classes, with LLM-refined reviews remaining close to human-written ones. This supports our core hypothesis: polishing a review with an LLM preserves the original human ideas.

Textual features, by contrast, reveal an interesting pattern. Entropy shows that LLM-refined reviews occupy an intermediate position between the two classes: closer to AI-generated text than to human, yet still somewhat distinguishable from both. This explains why, in our ablation study at Appendix B.6, textual features alone outperform semantic features for three-class classification.

B.5.2. CLAIM-LEVEL INTERPRETATION OF SEMANTIC OVERLAP

By examining Figure 19a, we confirm that AI-generated reviews exhibit higher semantic similarity to AI references than human-written ones. But what does this overlap look like in practice? To answer this, we present an illustrative example for Mean Best-Match Claim Similarity, the most discriminative feature identified by the classifier.

*Table 11.* Example of a target AI-generated review with high semantic overlap with AI reference reviews. Highlighted claims contribute most strongly to this overlap, while the remaining claims show lower semantic similarity.

---

**Target Review (DeepSeek-V3.1) Exhibiting High Semantic Overlap with Other AI Reviews**

(1) The derived policy gradient theorems for off-policy learning are rigorous and provide exact gradients.

(2) The paper demonstrates promising results in training policies without environment interaction.

The paper acknowledges scalability concerns but does not thoroughly address them.

Results vary significantly across environments, suggesting task-dependent effectiveness.

(3) The dimensionality of PVFs grows with policy parameters, which could hinder the performance of large policies.

Baselines like SAC or PPO are not included, and the paper focuses mainly on ARS and DDPG.

(4) How might PVFs scale to policies with millions of parameters (e.g., in modern deep RL)?

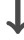

---

**Closest-Matching Claims from the AI-Generated Reference Reviews**

(1) The derivation of policy gradient theorems for both stochastic and deterministic cases is rigorous and well-presented, with clear algorithmic instantiations. ⇒ Qwen-3-235BA22B

(2) The zero-shot learning and offline learning experiments are compelling, showing the ability to train entirely new policies from scratch using a frozen PVF. ⇒ Gemini-2.5 Flash

(3) A major concern with PVFs is the curse of dimensionality with respect to the policy parameters theta. ⇒ Gemini-2.5 Pro

(4) Addressing the scalability of PVFs for very large policy networks, either through learned embeddings or other dimensionality reduction techniques, would substantially elevate the work. ⇒ Gemini-2.5 Flash

---

Table 11 shows an AI-generated review of an ICLR 2021 paper, shortened for clarity. While no single model matches all claims, together they provide broad coverage of the target review's points. This overlap produces a Mean Best-Match Claim Similarity of 0.8269, which is far above the 0.637 average observed for human reviews of the same paper.

### B.6. Feature Type Selection: Textual vs. Semantic vs. Combined

A core premise of our work is that robust three-class classification requires moving beyond purely textual or purely semantic features. Here, we provide empirical evidence supporting this design choice.

We trained three variants of our classifier: one using only the four textual features, another using only the five semantic features, and a third combining both sets: which constitutes Sem-Detect.

Figure 20 presents the results. The combined approach achieves a Macro-F1 score of approximately 0.84, outperforming both the textual-only variant (0.76) and the semantic-only variant (0.59). This performance gap highlights why neither feature type alone is sufficient for reliable three-class detection.

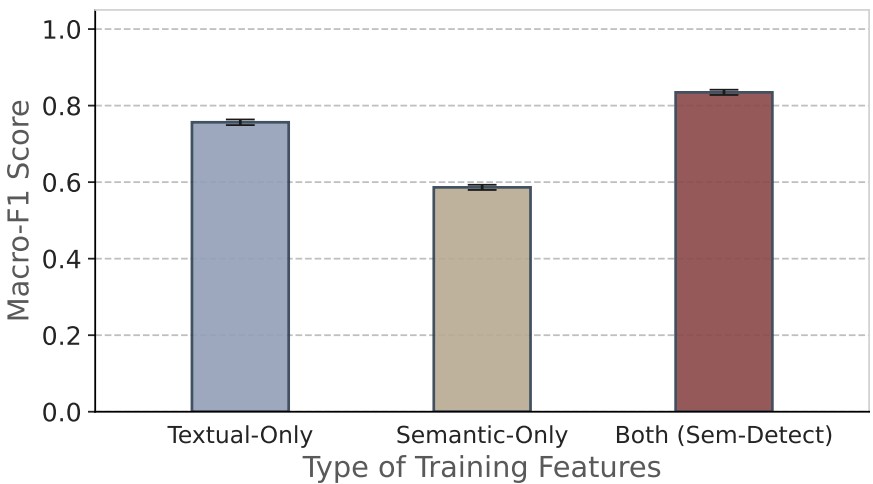

*Figure 20.* Impact of feature type on classification performance.

### B.7. Exhaustive Feature Subset Evaluation

The previous sections established that combining semantic and textual features is necessary for reliable three-class detection. A natural follow-up question is whether all nine features are needed, or whether a smaller subset could achieve comparable or even better performance. To answer this, we evaluate every possible feature combination: with nine features, there are $2^9 - 1 = 511$ non-empty subsets, and we test each one.

For each subset, we perform randomized hyperparameter search with five-fold stratified cross-validation, using Macro-F1 as the optimization target (the same protocol described in Appendix B.1) for selecting the final model.

*Table 12.* Selected results from the exhaustive evaluation of all 511 feature subsets. S = semantic features, T = textual features. Each subset is individually optimized via randomized hyperparameter search.

| Rank | Feature Subset | Types | Macro-F1 |
|------|----------------|-------|----------|
| 1 | Mean Best-Match Sim. \| Mean Pairwise Cos. Dist. \| Log-Review Length \| FastDetect \| Perplexity \| Entropy | S+T | 0.8416 |
| 2 | Mean Best-Match Sim. \| Mean Pairwise Cos. Dist. \| Log-Review Length \| Top-$k$ \| Entropy | S+T | 0.8401 |
| **18** | **All 9 features (Sem-Detect)** | **S+T** | **0.8354** |
| 334 | Top-$k$ \| FastDetect \| Perplexity \| Entropy *(best textual-only)* | T | 0.7317 |
| 453 | Mean Best-Match Sim. \| Mean Pairwise Cos. Dist. \| Log-Review Length *(best semantic-only)* | S | 0.6139 |
| 511 | Mean Pairwise Cos. Dist. alone *(worst)* | S | 0.3034 |

Table 12 summarizes the search space. Two patterns stand out. First, every top-300 subset mixes semantic and textual features: the best pure-textual combination ranks only 334th, and the best pure-semantic ranks 453rd, confirming that neither family alone is sufficient regardless of how features are combined. Second, performance among the top mixed subsets is tightly clustered: the gap between rank 1 (0.8416) and our full 9-feature model at rank 18 (0.8354) is just 0.0062, meaning the choice of which mixed subset to use matters far less than ensuring both types are present.

We further visualize the full search space in Figure 21, where we plot the Macro-F1 of every subset, grouped by whether it contains only semantic features, only textual features, or a mix of both. The separation is clear: mixed subsets occupy the upper region of the distribution, while pure-type subsets are concentrated in the lower ranks, with virtually no overlap between the two.

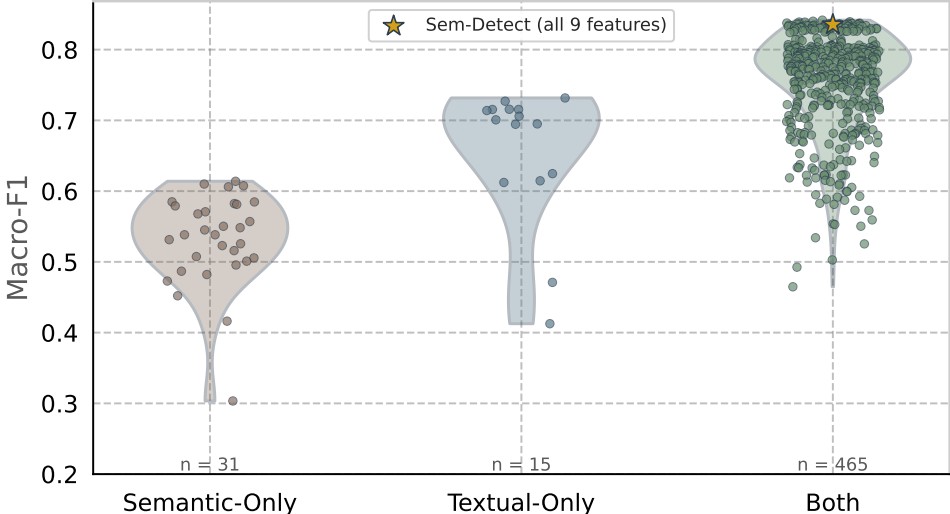

*Figure 21.* Distribution of Macro-F1 scores across all 511 feature subsets, grouped by feature composition: textual-only, semantic-only, and both (semantic + textual).

Together, these analyses provide comprehensive evidence that (i) the combination of semantic and textual features is structurally necessary and not an artifact of our particular selection, and (ii) the full feature set performs near-optimally within the space of possible subsets.

## C. Baseline Algorithm Details

We use the official implementations of all baseline detectors whenever they are available. In all cases, we follow the configurations and recommendations provided by the original authors.

For Anchor, we adopt the anchor-prompting strategy proposed by Yu et al. (2026). This approach requires a paper-specific prompt conditioned on the paper's content for each submission, which we generate using GPT-5 (Singh et al., 2025). We then tune the cosine-similarity threshold ($\theta$) on the training set at fixed TPR@% FPR values, and finally evaluate the method on the test set.

For EditLens, we use the authors' RoBERTa-Large model[5] to obtain the results in Figure 10. For the ICLR 2026 analysis in Section 5.6, we use the official predictions released by Pangram Labs and intersect them with our dataset to obtain EditLens scores for the overlapping reviews.

---

[5] https://huggingface.co/pangram/editlens_roberta-large

# D. Additional Details on Robustness to Generation Conditions

## D.1. Construction of Out-of-Distribution Evaluation Sets

As described in Section 3, although peer reviews follow a broadly shared structure, different AI conferences adopt distinct reviewing guidelines and templates. These differences may affect how AI-generated reviews are written, and therefore how well our method generalizes. To study this effect, we consider two OOD evaluation settings: OOD-M and OOD-M+P.

The OOD-M setting, where reviews are generated by unseen model families using the same prompt template as in training, is fully described in the main paper (Section 5.4). In this appendix, we therefore focus on the construction of the OOD-M+P setting, which introduces additional prompt variations not used during training.

**OOD-M+P evaluation.** We use the same three out-of-distribution models as in OOD-M: Claude-Sonnet-4, Mistral-Large-3, and GPT-oss-120B, but combine them with different prompt templates that differ from those used during training.

**Reviewer Personality Variations.** We define five distinct reviewer personalities that capture different reviewing styles commonly observed in academic peer review, and Table 13 presents the full personality prompts used in our experiments.

*Table 13.* Reviewer personality prompts used for generating AI reviews in the OOD-M+P setting.

| Personality | Prompt |
| --- | --- |
| Concise & Critical | You are a very concise reviewer. You like to go straight to the point and you mostly raise major issues with the work. Please be concise, critical, focused, and constructive so that the authors find the review convincing and improve their manuscript accordingly. |
| Detailed & Balanced | You are a thorough and balanced reviewer. You provide detailed analysis and carefully weigh both strengths and weaknesses of the work. Please be comprehensive, fair-minded, and constructive so that the authors find the review convincing and improve their manuscript accordingly. |
| Encouraging & Supportive | You are an encouraging and supportive reviewer. You tend to highlight the positive aspects of the work while gently pointing out areas for improvement. Please be positive, constructive, and motivating so that the authors find the review convincing and improve their manuscript accordingly. |
| Methodologically Rigorous | You are a methodologically rigorous reviewer. You focus heavily on experimental design, statistical validity, and reproducibility. Please be technically precise, methodical, and constructive so that the authors find the review convincing and improve their manuscript accordingly. |
| Novelty-Focused | You are a novelty-focused reviewer. You prioritize innovation and originality, and are particularly interested in how the work advances the field. Please be discerning about contributions, forward-thinking, and constructive so that the authors find the review convincing and improve their manuscript accordingly. |

**Main body and prompt combination.** In addition to reviewer personality, we also vary the structural format of the review. Specifically, we use three different main body templates: one matching the official ICLR 2021 reviewing guidelines, one matching the NeurIPS 2021 guidelines, and a general template suitable for ML conferences but distinct from our default prompt. For each review, we randomly select one reviewer personality and one main body template.

## D.2. Extended Analysis of Performance Under Distribution Shift

Table 14 presents the full class-wise performance of Sem-Detect under distribution shift for each class across three settings: In-Dist (same models and prompts as used in training), OOD-M (unseen models, same prompts), and OOD-M+P (unseen models and prompts).

*Table 14.* Class-wise performance under distribution shift.

| Class | In-Dist | | | OOD-M | | | OOD-M+P | | |
| --- | --- | --- | --- | --- | --- | --- | --- | --- | --- |
| | Prec. | Rec. | F1 | Prec. | Rec. | F1 | Prec. | Rec. | F1 |
| AI-Generated | 0.93 | 0.91 | 0.92 | 0.97 | 0.67 | 0.79 | 0.96 | 0.65 | 0.78 |
| LLM-Refined | 0.87 | 0.92 | 0.89 | 0.80 | 0.83 | 0.82 | 0.76 | 0.77 | 0.76 |
| Human | 0.75 | 0.64 | 0.69 | 0.45 | 0.63 | 0.53 | 0.42 | 0.63 | 0.50 |

The results reveal a consistent pattern across settings. AI-generated reviews maintain high precision (0.93-0.97) in all conditions, confirming that Sem-Detect's positive predictions for this class are reliable. The drop in AI recall under OOD (from 0.91 to 0.67 and 0.65) reflects conservative behavior: uncertain samples are routed away from the AI class rather than risking false accusations. LLM-refined performance remains stable under distribution shift, with both precision and recall staying above 0.76 across all settings. Human precision sees a moderate decrease, however, recall remains stable (0.63-0.64), indicating that the model continues to identify a majority of true human reviews.

### D.3. Comparison with Binary Baseline Detectors Under Distribution Shift

We extend the results of Section 5.4 by studying the impact of OOD data on baselines other than Sem-Detect. To enable comparison, we collapse the three classes into a binary setting and evaluate RADAR and Anchor under the same conditions.

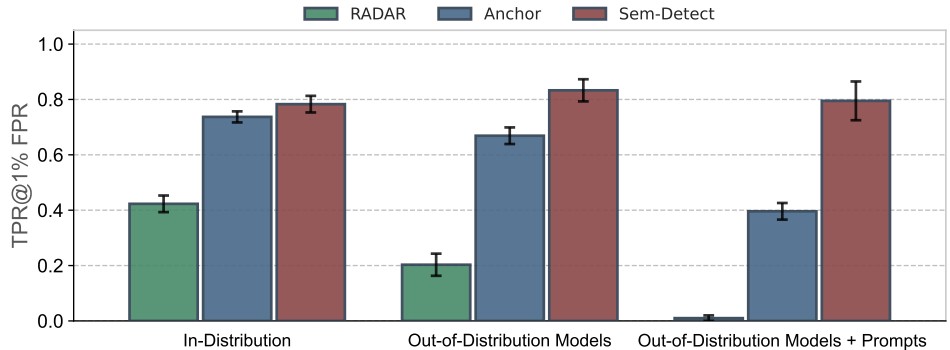

*Figure 22.* Binary generalization under distribution shift for Sem-Detect and baselines.

Figure 22 reveals that both baselines experience performance drops, but the most unexpected finding concerns RADAR. Unlike Sem-Detect, which explicitly uses the training models as reference points, RADAR was not optimized for any specific set of generators. From its perspective, reviews from training models should be no easier to classify than reviews from unseen ones. Yet RADAR shows the largest decline, with TPR at 1% FPR dropping substantially under OOD conditions. Anchor, by contrast, proves more stable, likely due to its reliance on semantic comparison rather than surface-level patterns. Sem-Detect, despite the performance drop observed in the three-class setting, maintains its TPR when evaluated from this binary perspective.

### D.4. Robustness to Reference Model Choice

The experiments in Section 5.4 evaluate Sem-Detect when the *target* reviews (i.e., those being classified) are produced by unseen models, while the reference reviews used for semantic comparison come from the same models as in training. In practice, however, a user deploying Sem-Detect may not have access to the specific LLMs used during training to generate reference reviews. We therefore study the complementary scenario: the target reviews come from models seen during training (Gemini-2.5-Pro, Qwen-3, and DeepSeek-V3.1), but the $k = 3$ reference reviews are produced by GPT-oss-120B, Mistral-Large-3, and Claude-Sonnet-4, which had no effect on the classifier training.

Table 15 shows that changing reference models (OOD-Ref) leads to a more modest degradation than changing target models (OOD-M): Macro-F1 drops from 0.84 to 0.79, compared to 0.71 under OOD-M. Most notably, AI recall remains at 0.92 under OOD-Ref versus 0.67 under OOD-M, while AI precision stays at 0.96. This suggests that the choice of reference models is less critical than the choice of target models for overall detection performance, and that users deploying Sem-Detect can substitute the reference models with whichever LLMs they have available, with only a moderate effect on overall performance.

*Table 15.* Comparison of distribution shift settings. OOD-M uses unseen target models with training reference models; OOD-Ref uses training target models with unseen reference models.

|  | In-Dist | OOD-M | OOD-Ref |
|---|---|---|---|
| Same Target Models | ✓ | ✗ | ✓ |
| Same Reference Models | ✓ | ✓ | ✗ |
| AI Precision | 0.93 | 0.97 | 0.96 |
| AI Recall | 0.91 | 0.67 | 0.92 |
| 3-Class Macro-F1 | 0.84 | 0.71 | 0.79 |

### D.5. Expanding the Training Generator Pool

The out-of-distribution results in Section 5.4 raise a natural question: would exposing Sem-Detect to a wider variety of generator families during training improve its performance under distribution shift? We investigate this through two experiments, both evaluated on the unchanged OOD-M+P test set. In each, we extend the original four-model generator set (Gemini-2.5-Flash, Gemini-2.5-Pro, DeepSeek-V3.1, and Qwen3-235B), with two new families: GLM-5 (GLM-5-Team, 2026) and Kimi-K2.5 (Kimi-Team, 2026).

*Table 16.* Effect of expanding the training generator pool on the OOD-M+P test set. Variant 1 uses all six families per paper. Variant 2 samples four of the six families per paper, keeping the original class distribution while exposing the classifier to all six families overall.

| Configuration | Macro-F1 | AI Prec. | AI Rec. |
|---|---|---|---|
| Original (Sem-Detect) | 0.68 | **0.96** | 0.65 |
| Variant 1: Full six-model pool | **0.70** | 0.88 | **0.73** |
| Variant 2: Per-paper subset | 0.69 | 0.89 | 0.72 |

**Variant 1: Full six-model pool.** We first use all six model families for every paper. This increases the number of AI and LLM-refined reviews and creates more training instances by allowing each target review to be paired with multiple non-overlapping sets of three reference reviews. As Table 16 shows, Macro-F1 increases from 0.68 to 0.70. However, this gain reflects a precision/recall trade-off: AI recall rises from 0.65 to 0.73, while AI precision drops from 0.96 to 0.88. Thus, the model detects more AI reviews, but also produces more false positives.

**Variant 2: Per-paper subset.** Variant 1 also changes the class balance, since adding two model families increases the number of AI and LLM-refined reviews relative to human reviews. To separate this effect from generator diversity, we run a second experiment where, for each paper, we randomly select four of the six model families. This keeps the original class distribution, sample counts, and $k = 3$ pairing scheme unchanged, while still exposing the classifier to all six generators across the dataset.

Variant 2 gives a Macro-F1 of 0.69, with AI precision of 0.89 and AI recall of 0.72. This closely matches Variant 1. Since the class distribution is now unchanged, the precision/recall trade-off cannot be explained by class imbalance; it is instead caused by broader generator exposure.

**Discussion.** Both variants produce only a small Macro-F1 gain but a large drop in AI precision. This is undesirable for our deployment setting, where falsely accusing a human reviewer is more costly than missing an AI-generated review. We therefore keep the original four-model configuration in the main paper. Still, the higher AI recall suggests that larger generator pools could be useful when combined with precision-preserving mechanisms, such as the confidence-based filtering in Section 5.3.

**D.6. Sensitivity to Partial AI Content**

In practice, a reviewer might selectively incorporate specific observations from an AI-generated review into their own assessment, instead of generating the final review end-to-end. We tested how Sem-Detect, despite not being trained for this setting, would behave under this scenario.

Starting from 90 papers in our test set, we construct synthetic hybrid reviews by systematically replacing human claims with AI-generated ones at controlled ratios. For each paper, we select the human review with the most substantive claims and a matching AI review (same evaluation score, different model family). We then use Claude-4.6-Opus (Anthropic, 2026) to replace 25%, 50%, or 75% of the human's evaluation, constructive input, and clarification claims with claims from the AI review, while preserving all factual restatements and meta-commentary from the original human review. The resulting mixed reviews are assembled to read as coherent single-author texts.

In total, we have five different types of contamination groups per paper: 0% (the original human review), 25%, 50%, 75%, and 100% (the source AI review), each with exactly 90 reviews. To avoid self-reference bias, the three AI reference reviews used for computing semantic features are drawn from model families that exclude the source AI review's author.

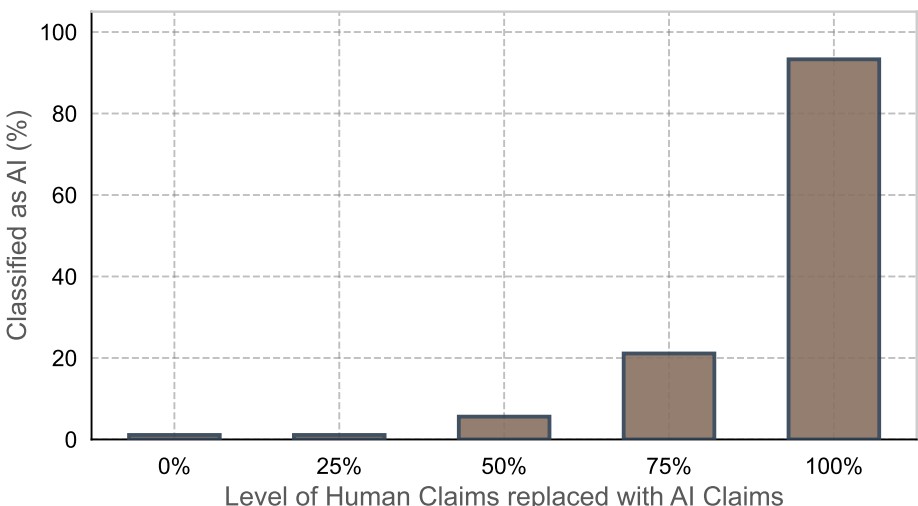

*Figure 23.* Fraction of reviews classified as AI-generated when human claims are increasingly replaced with AI-generated ones.

As shown in Figure 23, the number of reviews that are classified as AI-generated increases monotonically with contamination, confirming that Sem-Detect's semantic features are sensitive to the proportion of AI-originated claims. However, the full model remains conservative at low-to-moderate ratios, as the textual features (computed over the entire review text) anchor predictions toward the non-AI classes as the writing style is still predominantly human. The tipping point occurs at 75%, where the claim-level signal becomes strong enough to shift predictions substantially.

In the end, a reviewer who contributes genuine evaluative points alongside AI-suggested observations has, by definition, exercised human judgment over part of the review. Sem-Detect's decision boundary is not ambiguous in these cases; it correctly recognizes that human intellectual contribution is present and classifies accordingly. We note, however, that quantifying the precise degree of AI contamination within a single review is a distinct and complementary problem that falls outside the scope of this work.

# E. Factual Verification of Claims

Throughout this work, we have focused on modeling review authorship partially through the semantic content of expressed ideas. In particular, we examined whether these ideas are original, repetitive, or aligned with AI-generated references. However, this perspective captures only part of what makes a high-quality review. In fact, conference organizers have increasingly emphasized another essential aspect: factual accuracy. This growing concern is reflected in recent policy statements. For example, the ICLR 2026 Guidelines[6] explicitly state that "Reviews that feature false claims are a code of ethics violation." This raises the hypothesis on whether AI-generated reviews, might contain more factual errors than human-written ones, which could provide an additional detection signal.

To test this, we randomly sample 30 papers from ICLR 2021 and classify every extracted claim for factual accuracy using an LLM-as-a-judge pipeline (Gemini-3.0-Flash), incorporating two key considerations:

1. Not all claims are verifiable. Generic statements like "the document is well-written" cannot be checked against the paper content. We therefore fine-tune a BERT classifier to filter out such claims ($\approx$25% claims are discarded).

2. We target hallucinations rather than subjective assessment errors. Human reviewers may legitimately misinterpret aspects of a paper. Hallucinations, by contrast, are clear false statements that directly contradict the paper, for example, claiming "The method has not been evaluated on open-source LLMs" when there are experiments clearly reporting them. Our pipeline classifies each specific claim as either hallucinated or unverifiable.

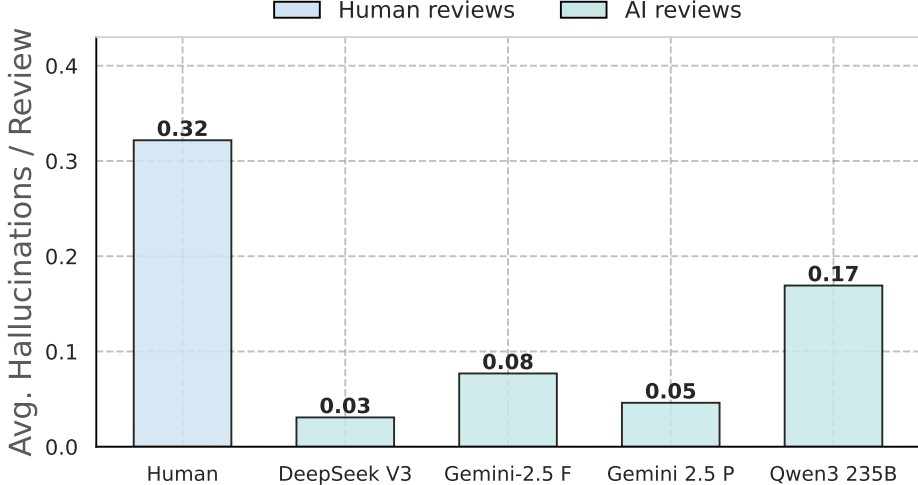

*Figure 24.* Average number of hallucinated claims per review across fully human and AI-generated reviews.

The results, shown in Figure 24, do not support our initial hypothesis. On average, human reviews contain 0.32 hallucinations per review, while all AI models produce fewer factual errors, ranging from 0.03 for DeepSeek-V3.1 to 0.17 for Qwen3-235B. Although the sample size is modest, manual verification indicates that the LLM-as-a-judge assessments are largely accurate.

One possible explanation is that AI models tend to be more conservative in their comments and engage with the paper at a more superficial level than human reviewers. As a result, they may be less likely to make specific factual claims and, therefore, less prone to hallucinations. This suggests that factual accuracy alone is not a reliable signal for distinguishing AI-generated reviews from human-written ones, as it could unfairly penalize careful human reviewers who simply avoid making mistakes.

Nevertheless, factual accuracy remains an important aspect of review quality and warrants further study. Future work could explore more advanced verification pipelines, for example by leveraging external document sources to validate factual claims more reliably.

---

[6]https://blog.iclr.cc/2025/11/19/iclr-2026-response-to-llm-generated-papers-and-reviews

Table 17 presents the LLM-as-a-Judge prompt for factual verification. We emphasize a conservative approach, flagging only clear hallucinations while treating misinterpretations or reasoning errors as acceptable.

*Table 17.* System and User Prompts used for Hallucination Detection in Peer Reviews.

---

### Hallucination Detection Prompt (Adapted for Brevity)

**System Prompt:**

You are an **extremely conservative hallucination detector for peer reviews**. You are given:

1. The full text of an academic paper.
2. The full peer review.
3. One claim extracted from the peer review.

Your task is to determine whether the reviewer has made a **HALLUCINATED claim** about the paper.

**Core Distinction**

You **must distinguish** between:

**Hallucination:** The reviewer asserts false content *as if it were explicitly stated in the paper*.

**Possible Incorrect Claim:** The reviewer may be wrong due to interpretation, reasoning, assumptions, or normative judgment. **These are not hallucinations and must be labeled `PASS.`**

If a claim could plausibly result from misunderstanding or reasoning, it **must be `PASS`**.

**What Counts as a Hallucination**

A claim is a hallucination **only if** the reviewer asserts an *objectively false fact* about the paper itself, like:

- Fabricated names, acronyms, or expansions
- Invented methods, stages, datasets, or definitions
- Claims that the paper introduces, defines, or uses something nonexistent
- Incorrect numeric values **only when**:
    - the number defines a *critical* component of the method or experiments, and
    - the reviewer presents it as a fact stated in the paper, and
    - the paper explicitly states a different value

**Label Definitions**

`CONTRADICTED`: Use **only** for clear hallucinations.

`PASS`: Use in **all other cases**, including possible incorrect claims.

---

**User Prompt:**

Paper Content: [parsed paper PDF]
Peer Review: [full review text]
Claim: [single extracted claim]

