# OpenReview forum: "Sem-Detect: Semantic Level Detection of AI Generated Peer-Reviews"
_ICML.cc/2026/Conference — ICML 2026 regular_

### Official Review · Reviewer_tiCW · 2026-03-12

**Soundness:** 2
**Presentation:** 3
**Significance:** 2
**Originality:** 3
**Overall Recommendation:** 3
**Confidence:** 3

**Summary:**

This paper proposes an AI-generated content detection method focusing on the semantic dimension. It rigorously compares the structural similarity of the target text with multiple AI reference samples at the claim level. Across a dataset of over 20,000 peer reviews from ICLR and NeurIPS conferences, Sem-Detect improves over the strongest prior detector by 36.5% in TPR@1% FPR in the binary setting

**Compliance With Llm Reviewing Policy:**

Affirmed.

**Final Justification:**

Thank you for the authors' reply and the mixed text experiment (W3) added during Rebuttal. I have carefully read your feedback and reassessed the contributions and limitations of this paper.

Based on the above clarification, I have decided to raise my score from Reject (2 points) to Weak Reject (3 points). However, I believe that this method still has inherent limitations in terms of the generalization and robustness of its underlying mechanisms:

- The biggest contribution of this paper is "shifting from text-level to semantic-level detection," but in the narrow domains that demand the deepest expertise, semantic features fail, forcing the system to degenerate and rely on traditional text surface features. This makes the method vulnerable to high-order adversarial attacks (such as deep rewriting).

- The supplementary experimental data shows that the system only begins to label AI on a large scale (21.1%) when the AI-generated substantive claims exceed 75%. This means that the tool is currently still a coarse-grained filter and cannot provide explanatory fine-grained attribution for the most common deep collaborative scenarios in academia, which involve "human-led core ideas + AI-assisted long logical chain derivation (accounting for approximately 30%-50%)".

- Because "building a general prior library will fail due to statistical necessity (Q1)," this means that the algorithm cannot be extended to real-time multi-turn dialogues, open-domain question answering, or domains lacking large-scale open-source review libraries (such as OpenReview).

**Key Questions For Authors:**

Is it possible to pre-build or continuously update a domain-specific "semantic claim prior library" using existing data to completely eliminate the computational reliance on real-time reference text generation?

Can this sensitive detection mechanism for homogeneous semantics be transferred to multi-agent collaborative networks for real-time filtering of redundant proposals and accelerating system convergence?

When detecting robot multimodal interaction logs or visual understanding feedback chains, how should the segmentation and feature extraction logic based on pure text claims be reconstructed?

**Limitations:**

The experiment and underlying mechanisms are highly tied to structured academic peer review scenarios, which may be limited in the detection of highly unstructured multi-turn open-domain dialogues or multimodal instruction generation.

**Strengths And Weaknesses:**

**Strengths**

This work successfully distinguishes between review texts written entirely by humans, human-written texts polished by AI tools, and review texts generated end-to-end by a large model.

It shifts the detection focus from the vulnerable "text surface style" to "underlying opinion attribution," protecting the valuable tool value of AI for researchers to reasonably utilize and enhance language expression.

It demonstrates stronger out-of-domain robustness than traditional statistical features and adversarial training methods in adversarial environments such as cross-model generation and prompt word manipulation.

Faced with adversarial environments such as cross-model generation and prompt word manipulation, it demonstrates stronger extra-domain robustness than traditional statistical features and adversarial training methods.

**Weaknesses**

However, each detection of a single text requires the real-time parallel generation of multiple AI reference texts on the same topic, resulting in excessive inference overhead and potentially making it unsuitable for real-time filtering of massive amounts of data.

The "claim extraction" step heavily relies on the parsing model's ability to follow instructions. For formula-intensive theoretical derivations or long logical chains, it is prone to extraction failure and disruption of the global contextual semantics.

When faced with long texts that combine core human innovations with deep AI logical deductions, the classifier's decision boundaries remain ambiguous and lack explanatory power.

The core features relied upon (i.e., the AI ​​model often arrives at similar conclusions) become completely ineffective in extremely narrow research directions with unique solutions, because human experts' conclusions also tend to converge significantly.

---

> ### Author Rebuttal · Authors · 2026-03-31
>
> Dear Reviewer,
>
> We sincerely appreciate the time and effort dedicated to our work. Below, we address each concern raised.
>
> ---
>
> **W1. Inference overhead from real-time reference generation**
>
> We would like to clarify several aspects that reduce overhead in practice. First, reference reviews are generated per paper, not per review. Second, detection does not need to happen in real time; Program Chairs (PC) can run Sem-Detect in batches after reviewing deadlines, a natural fit for how conferences operate. Batch APIs (already used; Appendix A.5) reduce LLM costs by up to 50%. Third, deployment costs are proportionally lower than training costs: at inference, we only generate AI references per paper and run claim extraction (no LLM-refined reviews are needed).
>
> From an individual user's perspective, we also measured end-to-end single-paper latency and cost, which came at under 2 minutes and <0.20$. We agree that Sem-Detect is not designed for large continuous real-time deployment, but this scenario does not arise in peer review. Conferences follow structured timelines with known deadlines, and PCs have full discretion over when to assess review quality. Running Sem-Detect in batches is both operationally realistic and fully supported by our pipeline's architecture.
>
> **W2. Claim extraction fragility for formula-intensive or long logical chains**
>
> The practical risk is lower than it may appear. Claim extraction operates on review text, not on the paper, so the context is constrained. While papers may contain dense notation, reviews are predominantly natural language. When reviewers reference equations, they do so briefly, and the extraction model only needs to copy such content. We validated that even a 32B model produces adequate extractions; we use Gemini-2.5-Flash for accessibility, not because the task demands a frontier model.
>
> **W3. Decision boundaries for reviews mixing human and AI content**
>
> We conducted a dedicated experiment. Using 90 test-set papers, we had Claude-4.6-Opus create hybrid reviews by replacing human substantive claims with AI-generated ones at controlled ratios (25/50/75%), preserving human factual restatements and meta-commentary.
>
> | AI Claim Ratio | Classified as AI (%) |
> | --- | --- |
> | 0% | 1.1 |
> | 25% | 1.1 |
> | 50% | 5.6 |
> | 75% | 21.1 |
> | 100% | 93.3 |
>
> The AI classification rate increases monotonically with contamination. At ≤50%, the model classifies most reviews as non-AI, since textual features reflect the predominantly human writing style. Still, we view this pattern as correct, as a review where a human contributes ~50% of the substantive evaluation has, nonetheless, substantial human intellectual contribution. The tipping point at 75% (where 21.1% of reviews are flagged) reflects the point at which AI-originated content begins to dominate the review's substantive claims.
>
> We acknowledge that quantifying the *precise degree* of AI contamination within a single review is a distinct and complementary problem that falls outside the current scope of Sem-Detect. We will include this experiment in the appendix.
>
> **W4. Convergence in narrow research directions**
>
> This is precisely why Sem-Detect combines semantic and textual features. As shown in Appendix B.5-Figure 17, Sem-Detect with semantic-only features achieves Macro-F1 of 0.59, textual-only reach 0.76, and the combination yields 0.84. Semantic features primarily separate fully AI-generated reviews (Figure 16a), while textual features distinguish LLM-refined from human writing (Figure 16b). In a scenario where semantic convergence reduces the reliability of one signal, the textual features, which are independent of claim overlap, continue to contribute.
>
> **Q1. Suggestion of a pre-built "semantic claim prior library":**
>
> We believe paper-specific reference generation is an architectural necessity. Sem-Detect asks: "Does this review say similar things to what AI models say *about this specific paper*?" A generic library would change this to: "Does this review resemble anything AI has ever said about *any* paper?" As such a library grows, every review finds a match by statistical inevitability, and the discriminative signal disappears.
>
> **Q2-3 Transfer to filtering redundant proposals and multimodal logs**
>
> Sem-Detect's core principle extends naturally to settings where expert evaluation must be authenticated (e.g., funding proposal reviews), requiring only domain-appropriate reference generation. The main barrier is data availability, as other domains lack public review repositories like OpenReview. Regarding multimodal content, reviewers typically discuss visual elements in natural language, which is captured during claim extraction. If a review depended entirely on interpreting visuals inaccessible to reference LLMs, semantic feature quality could degrade, but we consider this an edge case.
>
> ---
>
> **Conclusion:**
>
> We hope that our answers have addressed your concerns, and thank you once again for your valuable feedback.

---

> > ### Author Rebuttal · Reviewer_tiCW · 2026-04-03
> >
> > Thank you for the authors' reply and the mixed text experiment (W3) added during Rebuttal. I have carefully read your feedback and reassessed the contributions and limitations of this paper.
> >
> > Based on the above clarification, I have decided to raise my score from Reject to Weak Reject. However, I believe that this method still has inherent limitations in terms of the generalization and robustness of its underlying mechanisms:
> >
> > - The biggest contribution of this paper is "shifting from text-level to semantic-level detection," but in the narrow domains that demand the deepest expertise, semantic features fail, forcing the system to degenerate and rely on traditional text surface features. This makes the method vulnerable to high-order adversarial attacks (such as deep rewriting).
> >
> > - The supplementary experimental data shows that the system only begins to label AI on a large scale (21.1%) when the AI-generated substantive claims exceed 75%. This means that the tool is currently still a coarse-grained filter and cannot provide explanatory fine-grained attribution for the most common deep collaborative scenarios in academia, which involve "human-led core ideas + AI-assisted long logical chain derivation (accounting for approximately 30%-50%)".
> >
> > - Because "building a general prior library will fail due to statistical necessity (Q1)," this means that the algorithm cannot be extended to real-time multi-turn dialogues, open-domain question answering, or domains lacking large-scale open-source review libraries (such as OpenReview).

---

> > > ### Author Response · Authors · 2026-04-06
> > >
> > > Dear Reviewer,
> > >
> > > We sincerely thank you for taking the time to carefully review our rebuttal. We appreciate the score increase, and below we would like to provide further clarification on your remaining points.
> > >
> > > - **Semantic feature degradation in narrow domains makes method susceptible to deep rewriting (Point 1):**
> > >
> > >     As discussed in our response to W4, semantic and textual features are designed to complement each other: when semantic features become less discriminative, textual features compensate. Regarding vulnerability to deep rewriting, this concern applies broadly to all current AI-text detectors, including those based purely on textual features. While our OOD-M+P experiments do not directly simulate pure adversarial rewriting, they do introduce substantial stylistic variation through unseen models and prompts. Under these conditions (Appendix E.3), RADAR (which relies purely on surface-level features) suffers the largest performance drop, while Sem-Detect maintains its TPR. Even in a hypothetical narrow domain where semantic features contribute less, retaining both feature types makes Sem-Detect strictly more robust than a purely textual detector.
> > >
> > >     | TPR@1%FPR | In-Dist | OOD-M | OOD-M+P |
> > >     | --- | --- | --- | --- |
> > >     | RADAR | 0.423 | 0.203 | 0.01 |
> > >     | Anchor | 0.737 | 0.669 | 0.396 |
> > >     | **Sem-Detect** | **0.783** | **0.833** | **0.795** |
> > >
> > > - **On the 75% contamination threshold (Point 2):**
> > >
> > >     This scenario, where a human provides core judgments and an AI develops the supporting argumentation, represents an intermediate category not currently in the scope of Sem-Detect. Our LLM-refined class is intentionally restricted to language polishing without new content generation, which does not capture this case. We consider modeling this as a promising direction for future work, potentially requiring a more granular annotation scheme that distinguishes between claim *origination* and claim *elaboration*.
> > >
> > >
> > > - **On generalization beyond peer review (Point 3):**
> > >
> > >     We acknowledge that Sem-Detect is designed as a domain-specific tool for peer-review integrity. The observation that it does not extend to real-time multi-turn dialogues or open-domain QA reflects a scope boundary, not a methodological limitation. Many effective detection tools are domain-specific (e.g., Anchor or EditLens), and we believe this is appropriate given the high stakes and distinct characteristics of peer review.
> > >
> > >     That said, as we discussed in our rebuttal (Q2-3), Sem-Detect's core principle does generalize to other expert evaluation settings (e.g., funding proposals) where reference generation is feasible and data availability permits.
> > >
> > >
> > > ---
> > >
> > > We hope these clarifications address your remaining concerns, and thank you again for your feedback.

---

### Official Review · Reviewer_4F9N · 2026-03-12

**Soundness:** 2
**Presentation:** 3
**Significance:** 2
**Originality:** 3
**Overall Recommendation:** 2
**Confidence:** 3

**Summary:**

This paper proposes a new evaluation benchmark for detecting AI-generated peer reviews. Its core claim is that AI-generated reviews are highly similar, whereas human reviews are more unique. The experimental results verify its effectiveness.

**Compliance With Llm Reviewing Policy:**

Affirmed.

**Final Justification:**

Thank you for your detailed response. However, the necessity of the nine features has not yet been fully validated, and the reported OOD results indicate a significant drop in performance when the prompt or model is changed. Therefore, I tend to maintain the original score.

**Key Questions For Authors:**

1. The paper designs five semantic features and four textual features. Please explain and validate the necessity of each feature.
2. Different prompt templates may yield vastly different results, so it is recommended to explore various prompt templates.
3. The paper suggests encouraging models to generate diverse expressions during AI evaluation, but this seems inconsistent with real-world scenarios where people typically attempt to conceal such traces.
4. Is Figure 7 the result of fine-tuning or zero-shot learning?

**Limitations:**

Yes

**Strengths And Weaknesses:**

Strengths
1. This paper investigates a practical and meaningful topic, detecting whether peer reviews are AI-generated.
2. This paper constructs a new evaluation benchmark.


Weaknesses
1. Some technical details and experiments need to be supplemented and optimized. For example, the necessity of each of the nine features and the performance of different prompt templates.

---

> ### Author Rebuttal · Authors · 2026-03-31
>
> Dear Reviewer,
>
> We appreciate the time invested in reviewing our paper. Below, we provide responses to your questions.
>
> ---
>
> **W1. / Q1. The paper designs five semantic features and four textual features. Please explain and validate the necessity of each feature.**
>
> Feature relevance was something we carefully considered during development, and we provide a detailed analysis in the Appendices.
>
>  In Appendix B.4.1, we share the intuition behind each feature and report their importance scores (Figure 15), which show that no single feature disproportionately dominates the classification: each contributes meaningful signal. The distributional analysis in Figure 16 further shows that different features capture clearly different characteristics: semantic features like mean best-match claim similarity primarily separate AI-generated reviews from the other two classes, while textual features like entropy reveal that LLM-refined reviews occupy an intermediate position between human and AI text. For potentially redundant features such as entropy and perplexity, we empirically verified that including both leads to improved performance, as they capture complementary aspects of the language model's behavior.
>
> As for the necessity of combining both feature types, the ablation in Appendix B.5 (Figure 17) provides the strongest evidence: textual-only features achieve a Macro-F1 of ≈ 0.76, semantic-only ≈ 0.59, while the full combination reaches ≈ 0.84. Neither set alone is sufficient for reliable three-class detection. We will add forward references in the main text to make this analysis easier to find.
>
> **Q2. Different prompt templates may yield vastly different results, so it is recommended to explore various prompt templates.**
>
> We completely agree that this is relevant, and it is something we explicitly study. In Section 5.3, we evaluate Sem-Detect under two out-of-distribution settings: OOD-M (unseen model families, same prompt) and OOD-M+P (unseen model families *and* unseen prompts). The OOD-M+P setting combines 5 distinct reviewer personalities with 3 structural templates, yielding 15 prompt configurations not seen during training (Appendix E.1, Table 12). Under these conditions, we saw that detection is relatively robust to the input prompt and that AI Precision does not suffer from such modification. We will make this contribution more prominent in the updated version.
>
> |  | In-Dist | OOD-M | OOD-M+P |
> | --- | --- | --- | --- |
> | Same Target Models | ✓ | ✗ | ✓ |
> | Same Reference Models | ✓ | ✓ | ✗ |
> | AI Precision | 0.93 | 0.97 | 0.96 |
> | AI Recall | 0.91 | 0.67 | 0.65 |
> | 3-Class Macro-F1 | 0.84 | 0.71 | 0.68 |
>
> **Q3. The paper suggests encouraging models to generate diverse expressions during AI evaluation, but this seems inconsistent with real-world scenarios where people typically attempt to conceal such traces.**
>
> We would like to clarify that  we do *not* instruct models to produce creative or diverse reviews. On the contrary, as disclosed in Appendix A.2 (Table 4), we prompt models to produce critical, focused, and constructive reviews, which is likely how someone using an LLM to generate a review would actually prompt it. The key insight of Sem-Detect is that even under these realistic conditions, different AI models *naturally converge* on similar claims when reviewing the same paper, while human reviewers introduce more unique judgments. This convergence is not something we engineer; it is an empirical property we exploit. Asking models to generate deliberately diverse expressions would correspond more to an adversarial attack scenario, one that would also undermine the user's goal of producing a good review.
>
> **Q4. Is Figure 7 the result of fine-tuning or zero-shot learning?**
>
> Sem-Detect is trained only once on ML conference data (ICLR and NeurIPS). When we evaluate on MIDL, the model has not been exposed to any medical imaging reviews during training and no retraining is performed: this is what we refer to by "No domain-specific retraining is performed" in the caption of Figure 7. We will clarify this further in the text to avoid ambiguity.
>
> ---
>
> **Conclusion:**
>
> We hope that our answers have addressed your concerns, and thank you once again for your valuable feedback.
>
> Please let us know if any further clarification or additional information is needed from our end.

---

> > ### Author Rebuttal · Reviewer_4F9N · 2026-04-03
> >
> > Thank you for your detailed response. However, the necessity of the nine features has not yet been fully validated, and the reported OOD results indicate a significant drop in performance when the prompt or model is changed. Therefore, I tend to maintain the original score.

---

> > > ### Author Response · Authors · 2026-04-04
> > >
> > > Dear Reviewer, thank you for the follow-up. In response to your remaining concerns, we conducted a new experiment and offer additional context on both points.
> > >
> > > ---
> > >
> > > **Feature necessity.** To complement the existing per-feature analyses (Appendix B.4.1, Figures 15–16) and the type-level ablation (Appendix B.5), we conducted an incremental feature addition experiment. We add features one at a time, starting with the semantic features and then the textual ones, and retrain the LightGBM classifier from scratch at each step. The results are shown below:
> > >
> > > | # of Features | Feature added | Macro-F1 | ΔF1 |
> > > | --- | --- | --- | --- |
> > > | 1 | Mean best-match similarity | 0.5231 | — |
> > > | 2 | + Proportion high-similarity claims | 0.5393 | +0.0162 |
> > > | 3 | + Mean similarities above threshold | 0.5455 | +0.0062 |
> > > | 4 | + Mean pairwise cosine distance | 0.5566 | +0.0111 |
> > > | 5 | + Log review length | 0.5814 | +0.0248 |
> > > | 6 | + Percentage top-k | 0.6792 | +0.0978 |
> > > | 7 | + FastDetect score | 0.7796 | +0.1004 |
> > > | 8 | + Perplexity | 0.8308 | +0.0512 |
> > > | 9 | + Entropy | 0.8354 | +0.0046 |
> > > | ————————— | ————————— | ————————— | ————————— |
> > > | 10 | + Count of "gpu" (control) | 0.8359 | +0.0005 |
> > >
> > > We see that each of our selected features contributes a meaningful improvement. To verify that these gains are not an artifact of increasing dimensionality, we added an uninformative control feature (count the frequency of the word "gpu" in the review), which has no expected correlation with authorship. This control results on a negligible gain of +0.0005, which is roughly 9× smaller than even the smallest contributing feature.
> > >
> > > **Prompt sensitivity.** We note that Section 5.3 evaluates Sem-Detect under 15 unseen prompt configurations across 3 unseen model families. While Macro-F1 drops from 0.84 to 0.68 as expected, AI precision increases to 0.96, meaning that when the system does flag a review as AI-generated under distribution shift, it is correct 96% of the time. Furthermore, Appendix E.3 shows that when evaluated in the binary setting under the same OOD conditions, Sem-Detect maintains its TPR@1% FPR while baselines such as RADAR experience substantially larger drops.
> > >
> > > ---
> > >
> > > We hope this additional evidence fully addresses both concerns, and we remain happy to discuss further if needed.

---

### Official Review · Reviewer_xZwU · 2026-03-12

**Soundness:** 3
**Presentation:** 3
**Significance:** 3
**Originality:** 3
**Overall Recommendation:** 5
**Confidence:** 4

**Summary:**

This paper proposes Sem-Detect, a semantic-level detection framework for AI-generated peer reviews. The authors point out that existing detection methods typically rely on shallow text statistical features (like perplexity), making it difficult to distinguish between "pure AI-generated" text and "human text polished by LLMs." Sem-Detect addresses this pain point through the following:

1. Claim-based semantic overlap: It introduces this as the core detection signal, leveraging the observation that different AI models tend to produce similar viewpoints when reviewing the same paper.
2. Large-scale dataset: It constructs a three-class dataset containing over 20,000 reviews from ICLR and NeurIPS, covering fully human, LLM-polished, and fully AI-generated data.
3. Hybrid feature classifier: It combines text features (e.g., perplexity) with claim-level semantic features, using LightGBM for the final classification.

The method outperforms state-of-the-art baselines (such as RADAR and Anchor) in both binary and ternary classification tasks and demonstrates strong robustness in out-of-distribution (OOD) settings.

**Compliance With Llm Reviewing Policy:**

Affirmed.

**Key Questions For Authors:**

1. How does performance change when the number of available reference AI reviews per paper decreases? Adding an ablation study on reference-set size would be very helpful.

2. Can you provide a specific case study showing one or two examples where Sem-Detect succeeds but a text-only detector fails? This would make the practical impact of the semantic features much clearer.

3. What is the approximate cost and latency for running the complete inference pipeline on a single review? Even rough estimates would help evaluate the method's real-world usability.

4. During deployment, if the model's confidence is low, do you recommend threshold-based rejection, manual review, or another mechanism?

**Limitations:**

No, the authors do not adequately discuss limitations. While the Impact Statement (Page 9) touches on future AI capabilities and reputational risks, the paper lacks a dedicated section addressing methodological limitations. Specifically:

1. No explicit discussion of technical constraints:
   - No analysis of the method’s performance under extreme data scarcity (e.g., new domains or rare paper topics where few AI reviews exist for comparison).
   - No quantification of computational costs for claim extraction (e.g., latency per review) beyond API cost estimates.

2. Ethical risks are underemphasized:
   - The false-positive rate (~5% for LLM-refined reviews on ICLR 2026 data) is mentioned but not contextualized as a critical ethical concern for reviewers’ careers.

Suggestions for improvement:
- Add a dedicated Limitations section to discuss:
  - Generalizability to non-CS fields (beyond medical imaging).
  - Computational overhead of claim extraction for real-time systems.
  - Mitigation strategies for false accusations (e.g., tiered confidence thresholds with human review for uncertain cases).
- Explicitly link false-positive rates to ethical risks in the **Impact Statement**, rather than treating it as a minor practical detail.

This ensures reviewers can assess both the method’s capabilities and its responsible deployment.

**Strengths And Weaknesses:**

Strengths

A) Highly relevant problem setting. The paper tackles a crucial but previously under-explored issue: with conferences increasingly allowing limited LLM use, the real question isn't "is there an AI trace?" but "did a human still write this?". Distinguishing between fully AI-generated reviews and LLM-assisted human reviews is vital for the peer review process.

B) The semantic-level approach is the paper's most valuable contribution. The core observation—that different AI models generate highly similar claims for the same paper, whereas human reviewers bring independent judgments and diverse focuses—is sharp. Shifting detection from surface-level stylometry to claim-level semantic overlap is a much more targeted design than relying on statistical signals like perplexity, and it perfectly fits the specific task of peer review.

C) The three-class setup offers more practical value than standard binary classification. Many existing works still stick to coarse "AI vs. Human" detection. Breaking this down into human, LLM-refined, and fully AI is important. It matches real-world usage and aligns well with the nuanced governance needs of academic conferences (allowing assistance, but banning ghostwriting).

D) Comprehensive experimental coverage. The authors don't just report in-distribution results; they examine OOD performance across different models and prompts, and include transfer tests on cross-domain review data. This makes the evaluation much more convincing than works that just optimize for a single, fixed benchmark.

Weaknesses

A) Strong reliance on same-paper reference reviews. Sem-Detect's core advantage relies on semantically comparing the target review against reference AI reviews of the same paper. From a deployment perspective, I'm not convinced this prerequisite is always easy to meet. If a paper has very few reference AI reviews, or if the reference samples don't match the distribution of the generative model actually used, how much does performance degrade? The paper doesn't clarify this sufficiently.

B) Relatively complex inference pipeline. Compared to standard text detectors, this method requires claim extraction, followed by semantic matching, and finally classification. While methodologically sound, this introduces higher inference costs and a more complex deployment chain. The paper focuses heavily on accuracy gains, but the discussion on latency, cost, and engineering feasibility needs to be more concrete.

C) It's difficult to judge how much of the final gain truly comes from semantic modeling. I generally agree that the semantic features are a key contribution. However, since the final classifier still incorporates text and statistical features, I'd like to see more direct evidence showing the extent to which the model benefits from claim-level reasoning, rather than just picking up residual surface-level patterns in LLM-refined texts.

D) The cost of false positives in real-world scenarios needs more careful discussion. In this application, false positives have severe consequences. Incorrectly flagging a human-written or lightly polished review as fully AI-generated could cause unnecessary disputes in real conference environments. Therefore, I think this system is better framed as an auxiliary screening or risk-alert tool, rather than an automated system that makes final judgments. The paper would be more complete if it explicitly discussed this.

---

> ### Author Rebuttal · Authors · 2026-03-31
>
> ---
>
> Dear Reviewer,
>
> We greatly appreciate the time and effort you invested in reviewing our paper. Below, we provide a response to your comments.
>
> ---
>
> **W1 Strong reliance on same-paper reference reviews / Q1 How does performance change when the number of available reference AI reviews decreases?**
>
> We want to clarify that the number of reference reviews is controlled by the user deploying Sem-Detect: it is not a constraint imposed by the data: it simply means the user chose not to (or could not) generate more. That said, understanding whether three is the optimal number versus fewer is a fair question. We conducted an ablation varying k from 1 to 3:
>
> |  | Macro F1 |
> | --- | --- |
> | k=1 | 0.8190 |
> | k=2 | 0.8304 |
> | k=3 | 0.8379 |
>
> F1 Macro performance improves monotonically with k, but even a single reference review (k=1) achieves a Macro-F1 of 0.819, demonstrating that Sem-Detect does not critically depend on having multiple references. We use k=3 in our main experiments as it offers the best performance, but these results suggest that users could train Sem-Detect with lower values of k for better cost/latency performance.
>
> **Q2 Can you provide a case study where Sem-Detect succeeds but a text-only detector fails?**
>
> Yes. In our supplementary material, we include Sem-Detect's predictions for an ICLR 2022 test paper. We have now cross-referenced these with baseline predictions. RADAR (the strongest text-only baseline) misclassifies Reviewer 1 as AI-generated (a severe false accusation that Sem-Detect rarely makes). Sem-Detect correctly classifies this review as human-written, illustrating the practical value of incorporating semantic features.
>
> **W2 The discussion on latency, cost, and engineering feasibility needs to be more concrete / Q3 Cost and latency for inference pipeline on a single review?**
>
> We agree that Sem-Detect introduces additional cost and latency due to its multi-step pipeline, but this complexity is justified by the additional gains in robustness (critical in real world scenarios, as you also mention in W4). Regarding costs, a detailed breakdown is provided in Appendix A.5.
>
> As for latency, much of the pipeline can be parallelized with batch processing at scale. In a small-scale deployments, where an individual user wishes to analyze the reviews for a single OpenReview submission, our pipeline also uses parallelized API requests across providers so that results can be obtained in under 2 minutes and for <$0.20/ paper.
>
> **W3 It's difficult to judge how much of the final gain truly comes from semantic modeling.**
>
> We agree this is an important question and address it directly in Appendix B.5, where we compare three classifier variants: textual features only, semantic features only, and both combined.
>
> | Feature Set | Textual Only | Semantic Only | Both (Sem-Detect) |
> | --- | --- | --- | --- |
> | Macro F1 | 0.76 | 0.59 | 0.84 |
>
> The results show that neither feature type alone is sufficient for reliable three-class detection. The concern that the classifier could be picking up residual surface-level patterns in LLM-refined texts is reasonable. However, we verify that adding semantic features yields an 8-point improvement in Macro F1, demonstrating that claim-level analysis provides complementary signal beyond what textual features capture.
>
> **W4 The cost of false positives in real-world scenarios needs more careful discussion. / Q4 During deployment, if the model's confidence is low, what do you recommend?**
>
> We agree that preventing false accusations is critical and that Sem-Detect should be viewed as a risk-alert tool rather than a final decision system, as stated in our Impact Statement. We would also highlight that, among all evaluated methods, Sem-Detect achieves the lowest false accusation rate (Table 1).
>
> To further mitigate risk, we recommend the confidence-based thresholding strategy (Appendix D). Rather than forcing predictions on all inputs, low-confidence cases (below threshold θ) are deferred to human review. At θ = 0.80, ~79% of reviews are automatically classified at 94.7% accuracy, while ~21% are routed for manual inspection. More importantly, the remaining errors are predominantly benign (human reviews labeled as LLM-refined). We believe this two-tier approach offers a practical deployment path that balances automation with safeguards against misclassification.
>
> **Extending Limitations Discussion.**
>
> - We appreciate this suggestion. While we discuss several limitations across the Impact Statement and appendices (e.g., computational costs in Appendix A.5 and false-positive mitigation in Appendix D), we acknowledge that consolidating these into a dedicated section would improve clarity.
>
> ---
>
> **Conclusion:**
>
> We hope that our answers have addressed your concerns, and thank you once again for your valuable feedback.
>
> Please let us know if any further clarification or additional information is needed from our end.

---

> > ### Author Rebuttal · Reviewer_xZwU · 2026-04-02
> >
> > Fully resolved

---

> > > ### Author Response · Authors · 2026-04-06
> > >
> > > Dear Reviewer, we appreciate your support of our work.
> > >
> > > We are glad that our rebuttal has fully resolved your concerns.

---

### Official Review · Reviewer_vLtj · 2026-03-24

**Soundness:** 3
**Presentation:** 3
**Significance:** 2
**Originality:** 3
**Overall Recommendation:** 4
**Confidence:** 4

**Summary:**

This paper studies authorship attribution for peer reviews, aiming to distinguish fully AI-generated reviews from human-written reviews that may have been polished with LLMs. The proposed method, Sem-Detect, goes beyond surface text features by modeling claim-level semantic overlap between a target review and multiple AI-generated reference reviews for the same paper, then combines these signals with standard textual features in a three-way classifier. Using a dataset of human, LLM-refined, and fully AI-generated reviews built from 800 papers, the paper reports strong gains over prior detectors, good three-class performance, a very low human-to-AI false accusation rate, and reasonable robustness to unseen generators and cross-domain transfer.

**Compliance With Llm Reviewing Policy:**

Affirmed.

**Key Questions For Authors:**

N/A

**Limitations:**

Yes

**Strengths And Weaknesses:**

Strengths:

The paper addresses an important and emerging question for peer review: distinguishing fully AI-generated reviews from human-authored reviews that may have been polished by LLMs. This is more relevant than standard binary AI-text detection.
The authors provided clear technical ideas with high interpretability by using claim-level semantic overlap with manuscript-conditioned AI reference reviews rather than relying solely on surface text features.
The reported three-class confusion matrix is strong overall, especially for the AI and LLM-refined classes. Authors also do not stop at in-distribution evaluation. They include OOD experiments with unseen models and prompts, which is important for this application.


- Weaknesses:

The evidence for the three-class claim is weaker on the human vs. LLM-refined boundary than the paper's framing suggests: Although the paper emphasizes low human-to-AI confusion, the actual confusion matrix shows that 35.38% of human reviews are classified as LLM-refined, which is the dominant source of error. Since one of the paper's main contributions is precisely to separate fully human from LLM-refined reviews, this is a substantial weakness in the central empirical claim, not just a deployment concern.
OOD generalization drops materially, especially in recall for the AI class.
The baseline comparison is narrower than ideal for the central claim: The paper compares against general-purpose and domain-specific detectors, but the main contribution is a manuscript-conditioned semantic detector. A stronger evaluation would include more direct baselines that use manuscript-conditioned full-review similarity or simpler claim-matching heuristics.

---

> ### Author Rebuttal · Authors · 2026-03-31
>
> Dear Reviewer,
>
> We greatly appreciate the time and effort you invested in reviewing our paper. Below, we provide clarification to your comments.
>
> ---
>
> **W1.1 On the Human→LLM-Refined confusion rate.** We acknowledge that 35.38% of human reviews are classified as LLM-refined, and we agree this is the dominant source of error. However, we believe the practical severity of this error is substantially lower than its numeric magnitude suggests, for three reasons:
>
> 1. Even from a standard binary perspective (Human vs. Non-Human), Sem-Detect remains the strongest method across all baselines (Figure 4). The three-class formulation is an *additional* capability, not a replacement for binary detection.
> 2. The error direction that matters most in practice is whether legitimate LLM-assisted reviews are falsely accused of being fully AI-generated. This is the high-stakes confusion that could unjustly penalize reviewers who use LLMs responsibly. On this front, Sem-Detect misclassifies only 3.44% of LLM-refined reviews as AI-generated (Figure 5). The Human→LLM-Refined confusion, while undesirable, is a *conservative* error: it does not result in false accusations of misconduct.
> 3. As detailed in Appendix D, confidence-based thresholding offers a practical deployment path that substantially mitigates this issue. At θ=0.80, overall accuracy rises to 94.7% while still classifying 79% of reviews automatically, and the Human→LLM-Refined misclassification rate drops considerably (Figure 19b). We will make this deployment strategy more prominent in the updated version.
>
> **W1.2 On OOD generalization.** We agree that AI recall drops under distribution shift (from 0.91 to 0.67/0.65). However, we would highlight two aspects of how the model degrades, and present new evidence.
>
> First, AI *precision* actually increases to 0.96-0.97 under such OOD conditions (Table 2), meaning that when Sem-Detect does flag a review as AI-generated, that prediction is reliable. The recall drop reflects the model routing uncertain OOD samples toward the LLM-refined class rather than risking false accusations (a desirable failure mode).
>
> Second, as shown in Appendix E.3 (Figure 20), when evaluated from a binary perspective under the same OOD conditions, Sem-Detect maintains its TPR@1% FPR. This suggests that the OOD degradation is concentrated in the finer-grained three-class distinction rather than in the core ability to separate AI from non-AI content.
>
> Additionally, during this review period we conducted a new experiment studying a complementary and also deployment-relevant OOD scenario: what happens when the *reference* models (those used to generate AI reviews) are swapped? This setting (OOD-Ref) reflects the case where a conference deploying Sem-Detect can choose which LLMs generate references but cannot control what models reviewers use. The results are encouraging: Macro-F1 drops only to 0.79 (vs. 0.84 from the In-Distribution experiments), meaning that users can substitute reference models with whichever LLMs they have available with only moderate performance impact. We will include this experiment in the revised paper (Table below).
>
> |  | In-Dist | OOD-M | OOD-Ref |
> | --- | --- | --- | --- |
> | Same Target Models | ✓ | ✗ | ✓ |
> | Same Reference Models | ✓ | ✓ | ✗ |
> | AI Precision | 0.93 | 0.97 | 0.96 |
> | AI Recall | 0.91 | 0.67 | 0.92 |
> | 3-Class Macro-F1 | 0.84 | 0.71 | 0.79 |
>
> **W2: Need for manuscript-conditioned baselines and simpler claim-matching heuristics**
>
> We believe the requested comparison is largely addressed by our existing evaluation.
>
> Anchor (Yu et al., 2026) is precisely a manuscript-conditioned full-review similarity baseline: it generates a synthetic AI review for the target paper and compares it to the candidate review via embedding-level cosine similarity. This is, to our knowledge, the most direct existing instantiation of the approach you are describing, and it is included in our experiments (Table 1, Figures 3-4, and Appendix E.3).
>
> What Sem-Detect changes relative to Anchor is the *granularity* of semantic comparison: from full-review embeddings to claim-level matching. We also ablate this axis directly in two places. In Appendix B.3, we compare claim-level segmentation against sentence-level chunking (a simpler, heuristic-based segmentation), showing that semantically meaningful segmentation yields substantially better performance. We further provide a qualitative analysis of *why* sentence-level chunking fails (Table 10: over-segmentation breaks semantic units).
>
> We will add cross-references between these sections in the revision to make this coverage more immediately visible to readers.
>
> ---
>
> **Conclusion:**
>
> We hope that our answers have addressed your concerns, and thank you once again for your valuable feedback.
>
> Please let us know if any further clarification or additional information is needed from our end.

---

> > ### Author Rebuttal · Reviewer_vLtj · 2026-04-05
> >
> > I acknowledge teh author response. I agree that the kinds of errors are such that they do not matter that much, and I also agree about the suggested deployment path. However, I still believe that the manuscript-conditioned full-review similarity baseline is not enough.

---

> > > ### Author Response · Authors · 2026-04-07
> > >
> > > Dear Reviewer,
> > >
> > > Thank you for confirming that our clarifications on error directionality and the deployment path were convincing.
> > >
> > > On the remaining point about adding more baselines: during the discussion period, the authors of EditLens (Thai et al., 2026) officially released their detection model, which was not yet available when we submitted (as noted in Appendix C). Like Anchor, EditLens is a domain-specific peer-review detector from ICLR 2026, but it takes a different approach: instead of comparing reviews via semantic similarity, it estimates the degree of AI editing on a continuous scale.
> > > We took this opportunity to run EditLens on our pipeline under the same conditions as all other baselines. The updated Table 1 is below:
> > >
> > > | Detector | AUC ↑ | TPR @ 0.1% ↑ | TPR @ 1% ↑ |
> > > | --- | --- | --- | --- |
> > > | LogRank | 0.576 ± 0.01 | 0.000 ± 0.00 | 0.001 ± 0.00 |
> > > | MAGE | 0.699 ± 0.01 | 0.000 ± 0.00 | 0.008 ± 0.01 |
> > > | Fast-DetectGPT | 0.699 ± 0.01 | 0.021 ± 0.01 | 0.062 ± 0.01 |
> > > | Binoculars | 0.751 ± 0.01 | 0.008 ± 0.01 | 0.062 ± 0.02 |
> > > | TF Model † | 0.926 ± 0.01 | 0.369 ± 0.06 | 0.553 ± 0.04 |
> > > | RADAR | 0.965 ± 0.00 | 0.153 ± 0.05 | 0.371 ± 0.06 |
> > > | Anchor † | 0.979 ± 0.00 | 0.541 ± 0.04 | 0.713 ± 0.07 |
> > > | EditLens † → **New Result** | 0.998 ± 0.00 | 0.606 ± 0.16 | 0.956 ± 0.02 |
> > > | Sem-Detect † | **0.999 ± 0.00** | **0.760 ± 0.11** | **0.973 ± 0.03** |
> > >
> > > † Domain-specific detectors trained or tuned on peer-review data.
> > >
> > > We observe that even with this new detector, Sem-Detect outperforms domain-specific baselines across all metrics. The evaluation now covers three distinct detection philosophies tailored to peer review: full-review semantic similarity (Anchor), continuous edit intensity estimation (EditLens), and claim-level semantic analysis (Sem-Detect). We will include this new detector in the updated version.
> > >
> > > ---
> > >
> > > We hope this addresses your concern and thank you once again for the valuable feedback.

---

### Decision · Program_Chairs · 2026-04-30

**Decision:**

Accept (regular)

**Comment:**

This paper addresses an important and timely problem: detecting AI-generated peer reviews. Instead of relying on surface-level textual features, it proposes a new approach based on claim-level semantic similarity.
The design, which compares a target review with multiple AI-generated reference reviews, is intuitive and well-motivated. The paper is technically solid overall, supported by large-scale experiments and additional evaluations under out-of-distribution settings.
Reviewers generally agreed on the strengths of the paper, including the well-motivated problem setting, the novelty of semantic-level detection, and the comprehensive evaluation. At the same time, several important concerns were raised. The authors addressed many of these concerns in the rebuttal by providing additional experiments and clarifications, which improved the overall confidence in the work.
In summary, this paper presents a clear and novel contribution with practical relevance, and it offers a useful step forward for the field. While some limitations and open questions remain, they appear addressable in future work.